# Towards Semi-Supervised Learning with Non-Random Missing Labels

## Abstract

Semi-supervised learning (SSL) tackles the label missing problem by enabling the effective usage of unlabeled data. While existing SSL methods focus on the traditional setting, a practical and challenging scenario called label Missing Not At Random (MNAR) is usually ignored. In MNAR, the labeled and unlabeled data fall into different class distributions resulting in biased label imputation, which deteriorates the performance of SSL models. In this work, class transition tracking based Pseudo-Rectifying Guidance (PRG) is devised for MNAR. We explore the class-level guidance information obtained by the Markov random walk, which is modeled on a dynamically created graph built over the class tracking matrix. PRG unifies the history information of each class transition caused by the pseudo-rectifying procedure to activate the model's enthusiasm for neglected classes, so as the quality of pseudo-labels on both popular classes and rare classes in MNAR could be improved. We show the superior performance of PRG across a variety of the MNAR scenarios, outperforming the latest SSL solutions by a large margin. Checkpoints and evaluation code are available at the anonymous link `https://anonymous.4open.science/r/PRG4SSL-MNAR-8DE2` while the source code will be available upon paper acceptance.

## 1 Introduction

Semi-supervised learning (SSL), which is in the ascendant, yields promising results in solving the shortage of large-scale labeled data (Chapelle et al., 2009; Zhou, 2021; Van Engelen & Hoos, 2020). Current prevailing SSL methods (Lee et al., 2013; Berthelot et al., 2020; Sohn et al., 2020; Tai et al., 2021; Zhang et al., 2021) utilize the model trained on the labeled data to impute pseudo-labels for the unlabeled data, thereby boosting the model performance. Although these methods have made exciting advances in SSL, they only work well in the conventional setting, *i.e.*, the labeled and unlabeled data fall into the same (balanced) class distribution. Once this setting is not guaranteed, the gap between the class distributions of the labeled and unlabeled data will lead to a significant accuracy drop of the pseudo-labels, resulting in strong confirmation bias (Arazo et al., 2019) which ultimately corrupts the performance of SSL models. The work in Hu et al. (2022) originally terms the scenario of the labeled and unlabeled data belonging to mismatched class distributions as label Missing Not At Random (**MNAR**) and proposes an unified

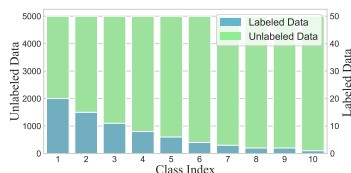

Figure 1: An example of the MNAR scenarios on CIFAR-10 (see Sec. 4 for details). The class distribution of total data is balanced whereas labeled data is unevenly distributed across classes. For better illustration, the y-axis has different scaling for labeled (blue) and unlabeled data (green).

doubly robust framework to train an unbiased SSL model in MNAR. It can be easily found that in MNAR, either the labeled or the unlabeled data has an imbalanced class distribution, otherwise, it degrades to the conventional SSL setting. A typical MNAR scenario is shown in Fig. 1, in which the popular classes of labeled data cause the model to ignore the rare classes, increasingly magnifying the bias in label imputation on the unlabeled data. It is worth noting that although some recent SSL methods (Kim et al., 2020; Wei et al., 2021) are proposed to deal with the class imbalance, they are still built upon the assumption of the matched class distributions between the labeled and unlabeled data, and their performance inevitably declines in MNAR.

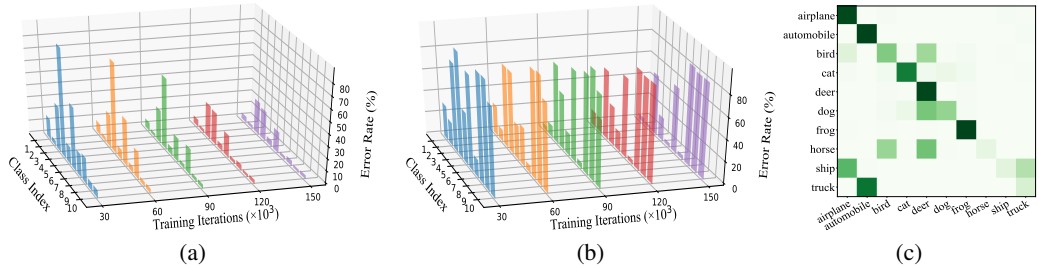

Figure 2: Results of FixMatch (Sohn et al., 2020) in MNAR and the conventional setting. The models are trained on CIFAR-10 with WRN-28-2 backbone (Zagoruyko & Komodakis, 2016). (a) and (b): Class-wise pseudo-label error rate. (c): Confusion matrix of pseudo-labels. In (b) and (c), experiments are conducted with the setting of Fig. 1, whereas in (a) with the conventional setting (*i.e.*, balanced labeled and unlabeled data). The label amount used in (a) is the same as that in (b) and (c).

MNAR is a more realistic scenario than the conventional SSL setting. In the practical labeling process, labeling all classes uniformly is usually not affordable because some classes are more difficult to recognize (Rosset et al., 2005; Misra et al., 2016; Colléony et al., 2017). Meanwhile, most automatic data collection methods also have difficulty in ensuring that the collected labeled data is balanced (Mahajan et al., 2018; Hu et al., 2022). In a nutshell, MNAR is almost inevitable in SSL. In MNAR, the tricky troublemaker is the mismatched class distributions between the labeled and unlabeled data. Training under MNAR, the model increasingly favors some classes, seriously affecting the *pseudo-rectifying* procedure. Pseudo-rectifying is defined as the change of the label assignment decision made by the SSL model for the same sample according to the knowledge learned at each new epoch. This process may cause *class transition*, *i.e.*, given a sample, its class prediction at the current epoch is different from that at the last epoch. In the self-training process of the SSL model driven by the labeled data, the model is expected to gradually rectify the pseudo-labels mispredicted for the unlabeled data in last epochs. With pseudo-rectifying, the model trapped in the learning of extremely noisy pseudo-labels will be rescued due to its ability to correct these labels.

Unfortunately, the pseudo-rectifying ability of the SSL model could be severely perturbed in MNAR. Take the setting in Fig. 1 for example. The model's "confidence" in predicting the pseudo-labels into the labeled rare classes is attenuated by over-learning the samples of the labeled popular classes. Thus, the model fails to rectify those pseudo-labels mispredicted as the popular classes to the correct rare classes (even if the class distribution is balanced in unlabeled data). As shown in Fig. 2b, compared with FixMatch (Sohn et al., 2020) trained in the conventional setting (Fig. 2a), FixMatch trained in MNAR (Fig. 1) significantly deteriorates its pseudo-rectifying ability. Even after many iterations, the error rates of the pseudo-labels predicted for labeled rare classes remain high. This phenomenon hints the necessity to provide additional guidance to the rectifying procedure to address MNAR. Meanwhile, as observed in Fig. 2c, we notice that the mispredicted pseudo-labels for each class are often concentrated in a few classes, rather than scattered across all other classes. Intuitively, a class can easily be confused with the classes similar to it. For example, as shown in Fig. 2c, the "automobile" samples are massively mispredicted as the most similar class: "truck". Inspired by this, we argue that it is feasible to guide pseudo-rectifying from the class-level, *i.e.*, pointing out the latent direction of class transition based on its current class prediction only. For instance, given a sample classified as "truck", the model could be given a chance to classify it as "automobile" sometimes, and vice versa. *Notably, our approach does not require predefined semantically similar classes.* We believe that two classes are conceptually similar only if they are frequently misclassified to each other by the classifier. In this sense, we develop a novel definition of the similarity of two classes, which is directly determined by model's output. Even if there are no semantically similar classes, as long as the model makes incorrect prediction during the training, this still leads to class transitions which has seldom been investigated before. Our intuition could be regarded as perturbations on some confident class predictions to preserve the pseudo-rectifying ability of the model. Such a strategy does not rely on the matched class distributions assumption and therefore is amenable to MNAR.

Given the motivations above, we propose class transition tracking based Pseudo-Rectifying Guidance (**PRG**) to address SSL in MNAR, which is shown in Fig. 3. Our main idea can be presented as

dynamically tracking the class transitions caused by pseudo-rectifying procedures at previous epoch to provide the class-level guidance for pseudo-rectifying at next epoch. We argue that every class transition of each pseudo-label could become the cure for the deterioration of the pseudo-rectifying ability of the traditional SSL methods in MNAR. A graph is first built on the class tracking matrix recording each pseudo-label's class transitions occurring in pseudo-rectifying procedure. Then we propose to model the class transition by the Markov random walk, which brings information about the difference in the propensity to rectify pseudo-labels of one class into various other classes. Specifically, we guide the class transitions of each pseudo-label during the rectifying process according to the transition probability corresponding to the current class prediction. The probability is obtained by the transition matrix of Markov random walk, which has been rescaled at both the class-level and the batch-level. Moreover, the class prediction at the last epoch can also be introduced to guide the pseudo-rectifying process at the current epoch. PRG recalls classes that are easily overlooked but appear in class transition history. They are deemed as similar to the ground-truth, and have more chance to be assigned rather than simply letting the model assign the classes it favors without hesitation. By this, PRG could help improve the quality of pseudo-labels suffered from biased imputation potentially caused by the mismatched distributions in MNAR. Because pseudo-rectifying is a spontaneous behavior of the model, moderately activating class transition will not hinder the learning of the model in the traditional setting. PRG is evaluated on several widely-used SSL classification benchmarks, demonstrating its effectiveness in coping with SSL in MNAR.

To help understand our paper, we summarize it with the following questions and answers.

- **What is the novelty and contribution?** Towards addressing SSL in MNAR, we propose transition tracking based Pseudo-Rectifying Guidance (**PRG**) to mitigate the adverse effects of mismatched distributions via combining information from the class transition history. We propose that the pseudo-rectifying guidance can be carried out from the class-level, by modeling the class transition of the pseudo-label as a Markov random walk on the graph.

- **Why does our method work for MNAR?** In MNAR, being aware of rare class plays a key role, PRG enhances the model to preserve a certain probability to generate class transition to rare classes when assigning pseudo-labels. This form of probability based on class transition history produces effective results, because we do not spare any attempt of the model to identify the rare class by class transition tracking (such attempts would be slowly buried due to overlearning of the popular classes). Thereby, PRG helps the model to still try to identify rare classes with a certain probability. This corresponds to our soft pseudo-label strategy, where we adjust the probability distribution of soft labels so that the model can assign pseudo-labels to rare classes with a clear purpose.

- **How about the performance improvement?** Our solution is computation and memory friendly without introducing additional network components. PRG achieves superior performance in the MNAR scenarios under various protocols, *e.g.*, it outperforms CADR (Hu et al., 2022), a newly-proposed method for addressing MNAR, by up to 15.11% in accuracy on CIFAR-10. Besides, we show the performance of PRG is also competitive in the traditional SSL setting.

## 2 RELATED WORK

*Semi-supervised learning* (SSL) is a promising paradigm to address the problem by effectively utilizing both labeled and unlabeled data. Given an input $x$ (labeled or unlabeled data), our objective in SSL can be described as the learning of a predictor for generating label $y$ for it. In conventional SSL settings (Berthelot et al., 2020; Sohn et al., 2020; Li et al., 2021; Zhang et al., 2021) and imbalanced SSL (Wei et al., 2021), underlying most of them is the assumption: labeled and unlabeled data are matched and balanced. Some more practical scenarios for SSL are now extensively discussed. Recently, some work has focused on addressing the class-imbalanced issue in SSL. Kim et al. (2020) refines the pseudo-labels softly by formulating a convex optimization. Wei et al. (2021) proposes class-rebalancing self-training combining *distribution alignment*. However, these existing methods still underestimate the complexity of practical scenarios of SSL, *e.g.*, Wei et al. (2021) works based on strong assumptions: the labeled data and unlabeled data fall in the same distribution (*i.e.*, their distributions match). Further, a novel and realistic setting called *label missing not at random* is proposed in Hu et al. (2022), which pops up in various fields such as social analysis, medical sciences and so on (Enders, 2010; Heckman, 1977). To address the mismatched distributions of labeled and unlabeled data in MNAR, Hu et al. (2022) proposes a class-aware doubly robust (CADR)

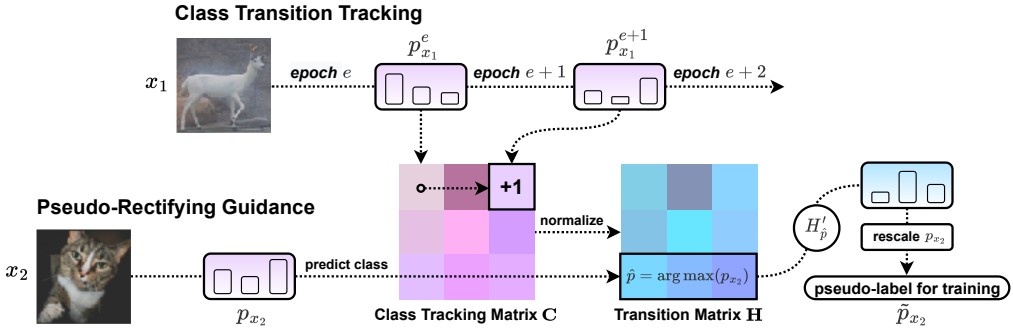

Figure 3: Overview of PRG. Class tracking matrix $\mathbf{C}$ is obtained by tracking the class transitions of pseudo-labels (*e.g.*, $p_{x_1}$ for sample $x_1$) between epoch $e$ and epoch $e+1$ caused by pseudo-rectifying procedure (Eq. (5)). The Markov random walk defined by transition matrix $\mathbf{H}$ (each row $H_i$ represents the transition probability vector corresponding to class $i$) is modeled on the graph constructed over $\mathbf{C}$. Generally, given a pseudo-label, *e.g.*, $p_{x_2}$ for sample $x_2$, class- and batch-rescaled $\mathbf{H}$ (*i.e.*, $\mathbf{H}'$) is utilized to provide the class-level pseudo-rectifying guidance for $p_{x_2}$ according to its class prediction $\hat{p} = \arg\max(p_{x_2})$ (Eqs. (6)~(7)). Finally, the rescaled pseudo-label $\tilde{p}_{x_2}$ is used for the training.

estimator combining class-aware propensity and class-aware imputation to remove the bias on label imputation. Differently, our method alleviates the bias from another perspective, that is to guide the pseudo-rectifying direction based on the historical information of class transitions.

## 3 METHOD

Formally, we denote the input space as $\mathcal{X}$ and the label space as $\mathcal{Y} = \{1, ..., k\}$ over $k$ classes. Following Hu et al. (2022), SSL can be reviewed as a label missing problem. The *label missing indicator* set is defined as $\mathcal{M}$ with $m \in \{0, 1\}$, where $m = 1$ indicates label is missing and $m = 0$ is the otherwise. Given the training dataset in SSL, we obtain a set of labeled data: $D_L \subseteq \mathcal{X} \times \mathcal{Y} \times \mathcal{M}$ and a set of unlabeled data: $D_U \subseteq \mathcal{X} \times \widehat{\mathcal{Y}} \times \mathcal{M}$. Since the ground-truth $y_U \in \widehat{\mathcal{Y}}$ of unlabeled data $x_U$ is inaccessible in SSL, prevailing self-training based SSL methods impute $y_U$ with pseudo-label $p$. $p = f(x_U; \theta)$ is predicted by the model which is parametrized by $\theta$ and trained on the labeled data. Let $(x_L^{(i)}, y_L^{(i)}, m_L^{(i)}) \in D_L, i \in \{1, ..., n_L\}$ be the labeled data pairs consisting of the sample with corresponding ground-truth label (*i.e.*, $m^{(i)} = 0$), and $(x_U^{(i)}, y_U^{(i)}, m_U^{(i)}) \in D_U, i \in \{n_L + 1, ..., n_T\}$ be the unlabeled data missing labels (*i.e.*, $m^{(i)} = 1$), where $n_L$ and $n_T$ refer to the number of labeled data and total training data respectively. Hereafter, the SSL dataset can be defined as $D = D_L \cup D_U$. In brief, we can review the conventional SSL as a optimization task for loss $\mathcal{L}$:

$$\min_\theta \sum_{(x,y,m) \in D} \mathcal{L}(x, y; \theta), \tag{1}$$

where $D$ is a dataset with independent $\mathcal{Y}$ and $\mathcal{M}$. In this sense, the model trained on $D_L$ can easily impute unbiased pseudo-labels for unlabeled data $x_U$ (Hu et al., 2022). Conversely, the scenario where $\mathcal{M}$ is dependent with $\mathcal{Y}$, namely label Missing Not At Random (MNAR), will make the model produce strong bias on label imputation, which causes the ability of pseudo-rectifying suffer greatly. Take the current most popular SSL method FixMatch (Sohn et al., 2020) as an example. In FixMatch, the term $\mathcal{L}(x, y; \theta)$ in Eq. (1) can be decomposed into two loss terms $\mathcal{L}_L$ and $\mathcal{L}_U$ with a pre-defined confidence threshold $\tau$ (implying $\max(p)$ is used as a measure of the model's confidence):

$$\mathcal{L}(x, y; \theta) = \mathcal{L}_L(x_L, y_L; \theta) + \lambda_U \mathbb{1}(\max(p) \geq \tau) \mathcal{L}_U(x_U, \arg\max(p); \theta), \tag{2}$$

where $\lambda_L$ is the unlabeled loss weight and $\mathbb{1}(\cdot)$ is the indicator function. Training with MNAR setting in Fig. 1, FixMatch is gradually seduced by samples predicted to be the labeled popular classes with confidence above $\tau$ (even though most of them are wrong), while samples predicted to be the rare class with confidence below $\tau$ do not participate into training, resulting in biased propensity on label imputation. In this work, we propose class transition tracking based Pseudo-Rectifying Guidance (**PRG**) to help model better self-correct pseudo-labels with additional guidance information.

### 3.1 PSEUDO-RECTIFYING GUIDANCE

Firstly, we formally describe the pseudo-rectifying process in SSL. In this paper, label assignment is considered as a procedure for generating soft labels. We denote the $i$-th component of vector $x$ as $x_i$. Let $p \in \mathbb{R}_+^k$ be the soft label vector assigned to unlabeled data $x_U$, where $\mathbb{R}_+$ is the set of nonnegative real numbers and $\sum_{i=1}^k p_i = 1$. Denoting $x$ at epoch $e$ as $x^e$, the pseudo-rectifying process can be described as the change on $p$ by the next epoch: $p^{e+1} = g_\theta(p^e)$, where $g_\theta(p^e)$ is a mapping from $p^e$ to $p^{e+1}$ determined by the knowledge learned from the model parametrized by $\theta$ at epoch $e+1$. In MNAR, take imbalanced $D_L$ and balanced $D_U$ as an example, as the training progresses, the model's confidence is gradually slashed and unexpectedly grows on the rare and popular classes in $D_L$ respectively. To address this issue, it is necessary to provide more guidance to assist the model in pseudo-rectifying. In general, the Pseudo-Rectifying Guidance (PRG) can be described as

$$\tilde{p}^{e+1} = \text{Normalize}(\eta \circ g_\theta(p^e)), \tag{3}$$

where $\circ$ is Hadamard product, scaling weight vector $\eta \in \mathbb{R}_+^k$ and $\text{Normalize}(x)_i = x_i / \sum_{j=1}^k x_j$.

We can review the technical contributions of some popular self-training works as obtaining more effective $\eta$ for pseudo-rectifying. For example, pseudo-labeling based methods (Lee et al., 2013; Sohn et al., 2020; Li et al., 2021; Xu et al., 2021; Zhang et al., 2021) set $\eta_i = 1/p_i^{e+1}, i \in \{i \mid i = \arg\max(p^{e+1}) \land p_i^{e+1} \geq \tau\}$ and $\eta_j = 0, j \in \{j \mid j \in (1, \cdots, k) \land j \neq i\}$ and, *i.e.*, using a confidence threshold to filter low-confidence samples. However, it is difficult to set an apposite $\eta$ at the sample-level (*e.g.*, for simplicity, Sohn et al. (2020) fixes $\tau$ to determine $\eta$ for all samples and the value of $\tau$ is usually set based on experience) to guide pseudo-rectifying, especially in the MNAR settings. In addition, some variants of class-balancing algorithms (Berthelot et al., 2020; Li et al., 2021; Gong et al., 2021) can be integrated into pseudo-rectifying framework. These methods utilize *distribution alignment* to make the class distribution of predictions close to the prior distribution (*e.g.*, the distribution of labeled data). This process can be summarized as dataset-level pseudo-rectifying guidance by setting $\eta$ as the ratio of the current class distribution of predictions to the prior distribution, *i.e.*, the fixed $\eta$ are used for all samples. Performing pseudo-rectifying guidance in this way strongly relies on an ideal assumptions: the labeled data and unlabeled data share the same class distribution, *i.e.*, in $D$, $\mathcal{Y}$ is independent with $\mathcal{M}$. Thus, these approaches fail miserably in the MNAR scenarios, which can be demonstrated in Appendix D.1. As we discussed in Sec. 1, it is also feasible to guide pseudo-rectifying at the class-level. Hence, we define rectifying weight matrix as $\mathbf{A} \in \mathbb{R}_+^{k \times k}$, where each row $A_i$ is representing the rectifying weight vector corresponding to class $i$. Denoting the class prediction as $\hat{p} = \arg\max(p)$, the class-level pseudo-rectifying guidance can be conducted by plugging $A_{\hat{p}^{e+1}}$ into $\eta$ in Eq. (3):

$$\tilde{p}^{e+1} = \text{Normalize}(A_{\hat{p}^{e+1}} \circ g_\theta(p^e)). \tag{4}$$

Next, we will introduce a simple and feasible way to obtain an effective $\mathbf{A}$ for PRG to improve the pseudo-labels predicted by SSL models in the MNAR scenarios.

### 3.2 CLASS TRANSITION TRACKING

Firstly, we consider building a fully connected graph $G$ in class space $\mathcal{Y}$. This graph is constructed by adjacency matrix $\mathbf{C} \in \mathbb{R}_+^{k \times k}$ (dubbed as class tracking matrix), where each element $C_{ij}$ represents the frequency of class transitions that occur from class $i$ to class $j$ (*i.e.*, an edge directed from vertex $i$ to vertex $j$ on $G$). $C_{ij}$ is parametrized by the following class transition tracking averaged on last $N_B$ batches with unlabeled data batch size $B_U$, *i.e.*, $C_{ij} = \sum_{n=1}^{N_B} C_{ij}^{(n)}/N_B$, where

$$C_{ij}^{(n)} = \left| \left\{ \hat{p}^{(b)} \mid \hat{p}^{(b),e} = i, \hat{p}^{(b),e+1} = j, i \neq j, b \in \{1, ..., B_U\} \right\} \right|, n \in \{1, ..., N_B\}, C_{ii}^{(n)} = 0. \tag{5}$$

Hereafter, we define the Markov random walk along the nodes of $G$, which is characterized by its transition matrix $\mathbf{H} \in \mathbb{R}_+^{k \times k}$. Each element $H_{ij}$ represents the transition probability for the class prediction $\hat{p}$ transits from class $i$ at epoch $e$ to class $j$ at epoch $e+1$. In specific, $\mathbf{H}$ is computed by conducting row-wise normalization on $\mathbf{C}$. The above designs are desirable for the following reasons.

(1) In the self-training process of the model, the historical information of pseudo-rectifying contains the relationship between classes, which is often ignored in previous methods and can be utilized

Figure 4: Visualization of class tracking matrix **C** obtained in training process of FixMatch (Sohn et al., 2020) on CIFAR-10 with the same setting as in Figs. 2c and 2b. The darker the color, the more frequent the class transitions. Overall, the number of class transitions decreases as the training progresses. Class transitions occur intensively between the popular classes, and class transitions between the rare classes gradually disappear (*e.g.*, between "ship" and "truck").

to help the model assign labels at a new epoch. We can record the class transition trend in pseudo-rectifying by Eq. (5), which corresponds to the transition probability represented by $H_{ij}$, *i.e.*, for a sample $x$, when its class prediction $\hat{p}$ is in the state of class $i$, if a rectifying procedure resulting in a class transition occurs, what probability will it transit to class $j$. Intuitively, given $p$ with $\hat{p} = i$, the model prefers to rectify it to another class similar to class $i$ in one class transition, *i.e.*, the preference of class transitions can also be regarded as the similarity between classes and the more similar two classes are, the more likely they are to be misclassified as each other's classes. The label is more likely to oscillate between the two classes, resulting in more swinging class transitions. As shown in Fig. 4, in the "dog" class predictions, the predictions transitioning to the "cat" class are significantly more than to other classes, and vice versa in the "cat" labels. We can observe that **C** behaves like a symmetric matrix, reflecting the symmetric nature of class similarity. Consequently, this similarity between classes can be utilized to provide information for our class-level pseudo-rectifying guidance.

(2) In the MNAR settings, the tricky problem is that the mismatched distributions lead to biased label imputation for unlabeled data. The feedback loop of self-reinforcing errors is not achieved overnight. Empirically, as the training progresses, the model becomes more and more confident in the popular classes (in labeled or unlabeled data), which leads to misclassify the samples that it initially thought might be the rare classes to the popular classes later. As shown in Fig. 4, the lower left corner and upper right corner of the heatmap (*i.e.*, the class transitions between the popular classes and rare classes) is getting lighter and always lighter than the upper left corner (*i.e.*, the class transitions among the popular classes), which means the model is increasingly reluctant to transfer the class prediction to the rare classes during the pseudo-rectifying process. If we only focus on what the model has learned at present, the model's past efforts to recognize the rare classes will be buried. The latent relational information between classes is hidden in the pseudo-rectifying process producing class transitions. The history of class transitions can point the way for bias removal on label imputation with an abnormal propensity on different classes caused by mismatched distributions in MNAR.

With obtained **H**, some preparations are done for plugging it into Eq. (4) to replace **A**. We're only modeling the pseudo-rectifying process resulting in class transition (*i.e.*, $C_{ii} = 0$), which means $H_{ii} = 0$, *i.e.*, $\eta_{\hat{p}^{e+1}}$ is set to 0 in Eq. (3). This will encourage the class prediction to transition to other classes during each pseudo-rectifying process, which is unreasonable for training a robust classifier. Hence, we control the probability that does not transition class by setting $H_{ii} = \frac{\alpha}{k-1}$, where $\frac{1}{k-1}$ is the average of the transition probabilities in each row of **H** and $\alpha$ is a pre-defined hyper-parameter. In addition, to avoid training instability, we scale each element in **H** by

$$H'_{ij} = \frac{\sum_{d=1}^{k} L_d}{\sum_{d=1}^{k} \sum_{d'=1}^{k} C_{dd'}} \times \frac{\sum_{d=1}^{k} C_{id}}{L_j} \times H_{ij}, \qquad (6)$$

where $L \in \mathbb{R}_+^k$ and $L_i$ records the number of class predictions belonging to class $i$ averaged on last $N_B$ batches. The first term on the right-hand side of Eq. (6) rescales $H_{ij}$ at the batch-level while the second term rescales $H_{ij}$ at the class-level, controlling the intensity of class transition together. For specific, the excessive class transitions in the self-training loop could yield confused supervision information that is not conducive to learning. Hereafter, to simply integrate our method into the above framework of pseudo-rectifying guidance, we plug **H**′ into **A** in Eq. (4):

$$\tilde{p}^{e+1} = \text{Normalize}(H'_{\hat{p}^{e+1}} \circ g_\theta(p^e)), \qquad (7)$$

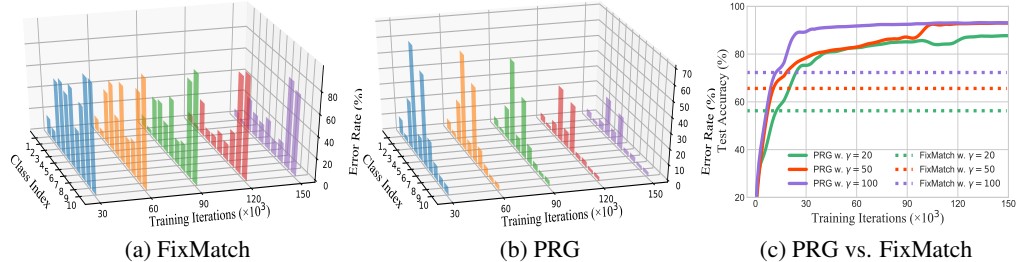

(a) FixMatch        (b) PRG        (c) PRG vs. FixMatch

Figure 5: Results on CIFAR-10 under CADR's protocol. (a) and (b): Class-wise pseudo-label error rate with $\gamma = 50$. (c): Learning curve of PRG. We mark the final results of FixMatch as dash lines.

Table 1: Mean accuracy (%) in MNAR under CADR's protocol. The results of baseline methods are derived from CADR (Hu et al., 2022). The larger $\gamma$, the more imbalanced the labeled data. In the format $\text{Mean}^{\uparrow\downarrow\texttt{Diff.}}_{\pm\texttt{Std.}}$, our accuracies are averaged on 3 runs while the standard deviations ($\pm\texttt{Std.}$) and the performance difference ($\uparrow\downarrow\texttt{Diff.}$) compared to FixMatch (Sohn et al., 2020) are reported.

| Method | CIFAR-10 | | | CIFAR-100 | | | mini-ImageNet | |
|---|---|---|---|---|---|---|---|---|
| | $\gamma = 20$ | 50 | 100 | 50 | 100 | 200 | 50 | 100 |
| Π Model | 21.59 | 27.54 | 30.39 | 24.95 | 29.93 | 33.91 | 11.77 | 15.30 |
| MixMatch | 26.63 | 31.28 | 28.02 | 37.82 | 41.32 | 42.92 | 13.12 | 18.30 |
| ReMixMatch | 41.84 | 38.44 | 38.20 | 42.45 | 39.71 | 39.22 | 22.64 | 23.50 |
| FixMatch | 56.26 | 65.61 | 72.28 | 50.51 | 48.82 | 50.62 | 23.56 | 26.57 |
| + Crest | $51.10^{\downarrow 5.16}$ | $55.40^{\downarrow 10.21}$ | $63.60^{\downarrow 8.68}$ | $40.30^{\downarrow 10.21}$ | $46.30^{\downarrow 2.52}$ | $49.60^{\downarrow 1.02}$ | – | – |
| + DARP | $63.14^{\uparrow 6.88}$ | $70.44^{\uparrow 4.83}$ | $74.74^{\uparrow 2.46}$ | $38.87^{\downarrow 11.64}$ | $40.49^{\downarrow 8.33}$ | $44.15^{\downarrow 6.47}$ | – | – |
| + CADR | $79.63^{\uparrow 23.37}$ | $93.79^{\uparrow 23.37}$ | $93.97^{\uparrow 21.69}$ | $59.53^{\uparrow 9.02}$ | $60.88^{\uparrow 12.06}$ | $63.30^{\uparrow 12.68}$ | $29.07^{\uparrow 5.51}$ | $32.78^{\uparrow 6.21}$ |
| + PRG (Ours) | $\mathbf{94.04}^{\uparrow 37.78}_{\pm 0.18}$ | $\mathbf{94.09}^{\uparrow 28.48}_{\pm 0.18}$ | $\mathbf{94.28}^{\uparrow 22.00}_{\pm 0.22}$ | $59.11^{\uparrow 8.60}_{\pm 0.54}$ | $61.84^{\uparrow 13.02}_{\pm 0.45}$ | $\mathbf{63.41}^{\uparrow 12.79}_{\pm 4.08}$ | $\mathbf{44.28}^{\uparrow 20.72}_{\pm 0.54}$ | $\mathbf{44.99}^{\uparrow 18.42}_{\pm 1.25}$ |
| + PRG$^{\text{Last}}$ (Ours) | $93.81^{\uparrow 37.55}_{\pm 0.98}$ | $93.44^{\uparrow 27.83}_{\pm 1.05}$ | $93.48^{\uparrow 21.20}_{\pm 0.79}$ | $\mathbf{59.54}^{\uparrow 9.03}_{\pm 0.99}$ | $\mathbf{62.36}^{\uparrow 13.54}_{\pm 0.23}$ | $60.56^{\uparrow 9.94}_{\pm 1.86}$ | $40.73^{\uparrow 17.17}_{\pm 1.27}$ | $43.89^{\uparrow 17.32}_{\pm 0.14}$ |

where $H'_{\hat{p}^{e+1}}$ can be regarded as the class prediction for one sample randomly walks along the nodes of $C$ at the current epoch, *i.e.*, drive a possible class transition in the pseudo-rectifying for bias removal on label imputation propensity due to MNAR (more discussions can be found in Appendix B). We note that it is also feasible to use the class transition driven by $\hat{p}^e$ to revise $p^{e+1}$ (what is the class prediction after a class transition from last epoch to the present), *i.e.*, replace $H'_{\hat{p}^{e+1}}$ in Eq. (7) with $H'_{\hat{p}^e}$, which is dubbed as **PRG$^{\text{Last}}$**. The whole algorithms are presented in Appendix A.

## 4 EXPERIMENT

**Dataset and Baselines.** We evaluate PRG on three widely used benchmarks in SSL, including CIFAR-10, CIFAR-100 (Krizhevsky et al., 2009) and mini-ImageNet (Vinyals et al., 2016) (a subset of ImageNet (Deng et al., 2009) composed of 100 classes). Following Hu et al. (2022), we mainly report the mean accuracy of PRG in both conventional SSL settings and various MNAR scenarios. Multiple baseline methods are compared, including representative conventional SSL algorithms: Π Model (Rasmus et al., 2015), MixMatch (Berthelot et al., 2019), ReMixMatch (Berthelot et al., 2020), and FixMatch (Sohn et al., 2020). More importantly, we provide fair comparisons with the recent label bias removal methods for imbalanced SSL: DARP (Kim et al., 2020), Crest (Wei et al., 2021), and the latest approaches designed for addressing SSL in MNAR: CADR (Hu et al., 2022).

**MNAR Settings.** Following Hu et al. (2022), the MNAR scenarios are mimicked by constructing the class-imbalanced subset of the original dataset for either the labeled data or the unlabeled data. Let $\gamma$ denote the imbalanced ratio, $N_i$ and $M_i$ respectively refer to the number of the labeled and the unlabeled data in class $i$ from $k$ classes. Three MNAR protocols are used for the evaluations on PRG: (1) CADR's protocol (Hu et al., 2022). $N_i = \gamma^{\frac{k-i}{k-1}}$, in which $N_1 = \gamma$ is the maximum number of labeled data in all classes, and the larger the value of $\gamma$, the more imbalanced the class distribution of the labeled data. For example, Fig. 1 shows CIFAR-10 with $\gamma = 20$. (2) Our protocol. Because the

Table 2: Mean accuracy (%) in MNAR under our protocol with the varying labeled data sizes $n_L$ and imbalanced ratios $N_1$. Baseline methods are based on our reimplementation.

| Method | CIFAR-10 ($n_L = 40$) | | CIFAR-10 ($n_L = 250$) | | CIFAR-100 ($n_L = 2500$) | | mini-ImageNet ($n_L = 1000$) | |
|---|---|---|---|---|---|---|---|---|
| | $N_1 = 10$ | 20 | 100 | 200 | 100 | 200 | 40 | 80 |
| FixMatch | $85.72_{\pm 0.93}$ | $76.53_{\pm 3.03}$ | $69.76_{\pm 5.57}$ | $46.53_{\pm 8.12}$ | $61.31_{\pm 3.67}$ | $41.38_{\pm 2.84}$ | $36.20_{\pm 0.36}$ | $28.33_{\pm 0.41}$ |
| + CADR | $85.54^{\downarrow 0.18}_{\pm 2.07}$ | $75.11^{\uparrow 1.42}_{\pm 3.41}$ | $92.25^{\uparrow 22.49}_{\pm 1.61}$ | $63.92^{\uparrow 17.39}_{\pm 5.47}$ | $\mathbf{61.62}^{\uparrow 0.31}_{\pm 0.93}$ | $46.16^{\uparrow 4.78}_{\pm 1.45}$ | $36.08^{\downarrow 0.12}_{\pm 0.84}$ | $30.52^{\uparrow 2.19}_{\pm 0.99}$ |
| + PRG (Ours) | $\mathbf{91.87}^{\uparrow 6.15}_{\pm 1.05}$ | $77.44^{\uparrow 0.91}_{\pm 15.96}$ | $\mathbf{93.93}^{\uparrow 24.17}_{\pm 0.16}$ | $\mathbf{67.86}^{\uparrow 21.33}_{\pm 16.98}$ | $61.49^{\uparrow 0.18}_{\pm 3.93}$ | $\mathbf{49.84}^{\uparrow 8.46}_{\pm 1.37}$ | $\mathbf{39.99}^{\uparrow 3.79}_{\pm 0.76}$ | $\mathbf{35.39}^{\uparrow 7.06}_{\pm 0.47}$ |
| + PRG$^{\text{Last}}$ (Ours) | $85.66^{\downarrow 0.06}_{\pm 5.93}$ | $\mathbf{77.85}^{\uparrow 1.32}_{\pm 1.86}$ | $92.80^{\uparrow 23.04}_{\pm 1.44}$ | $64.00^{\uparrow 17.47}_{\pm 5.02}$ | $60.41^{\downarrow 0.90}_{\pm 1.01}$ | $43.80^{\uparrow 2.42}_{\pm 1.71}$ | $39.84^{\uparrow 3.64}_{\pm 0.05}$ | $33.17^{\uparrow 4.84}_{\pm 0.52}$ |

total number of labeled data $n_L$ in the CADR's protocol varies with $\gamma$, which violates the principle of controlling variables, $n_L$ is fixed by users in our protocol. $N_1$ is altered for different scales of imbalance, i.e., $N_i = N_1 \times \gamma^{-\frac{i-1}{k-1}}$ while $\gamma$ is calculated by the constraint $\sum_{i=1}^{k} N_i = n_L$. We further consider the MNAR settings where the unlabeled data is also imbalanced, i.e., $M_i = M_1 \times \gamma_u^{-\frac{k-i}{k-1}}$ (implying inversely imbalanced distribution compared with the labeled data), where $M_1 = 5000$ in CIFAR-10. (3) DARP'sprotocol (Kim et al., 2020): $N_i = N_1 \times \gamma_l^{-\frac{i-1}{k-1}}$, $M_i = M_1 \times \gamma_u^{-\frac{i-1}{k-1}}$, where $N_1 = 1500$ and $M_1 = 3000$ in CIFAR-10, where $\gamma_l$ and $\gamma_u$ are varied for labeled and unlabeled data respectively, i.e., the distributions of the labeled and unlabeled data are mismatched and imbalanced.

**Implementation Details.** In this section, PRG is implemented as a plugin to FixMatch (Sohn et al., 2020) (see Appendix D.2 for other SSL learners). Thus, we keep the same training hyper-parameters as FixMatch (e.g., unlabeled data batch size $B_U = 448$), whereas the class invariance coefficient $\alpha = 1$ and the tracked batch number $N_B = 128$ are set for PRG in all experiments. The complete list of hyper-parameters can be found in Appendix C. Following Sohn et al. (2020), our models are trained for $2^{20}$ iterations, using the backbone of WideResNet-28-2 (WRN) (Zagoruyko & Komodakis, 2016) for CIFAR-10, WRN-28-8 for CIFAR-100 and ResNet-18 (He et al., 2016) for mini-Imagenet.

## 4.1 RESULTS IN MNAR

**Main Results.** The experimental results under CADR's and our protocol with various levels of imbalance are summarized in Tabs. 1 and 2. PRG consistently achieves higher accuracy than baseline methods across most of the settings, benefiting from the information offered by class transition tracking. As shown in Figs. 5a and 5b, the pseudo-rectifying ability of PRG is significantly improved compared with the original FixMatch, i.e., as the training progresses, the error rates of both the popular classes and the rare classes of the labeled data are greatly reduced, eventually

Table 3: Geometric mean scores (GM) on CIFAR-10 under CADR's protocol.

| Method | $\gamma = 20$ | $\gamma = 50$ | $\gamma = 100$ |
|---|---|---|---|
| FixMatch | $41.90_{\pm 8.55}$ | $53.61_{\pm 2.29}$ | $60.35_{\pm 1.84}$ |
| + CADR | $75.25^{\uparrow 33.35}_{\pm 1.55}$ | $92.98^{\uparrow 39.37}_{\pm 0.43}$ | $93.15^{\uparrow 32.8}_{\pm 0.36}$ |
| + PRG | $\mathbf{93.53}^{\uparrow 51.63}_{\pm 0.39}$ | $\mathbf{93.70}^{\uparrow 40.19}_{\pm 0.20}$ | $\mathbf{93.94}^{\uparrow 33.69}_{\pm 0.35}$ |
| + PRG$^{\text{Last}}$ | $93.35^{\uparrow 51.45}_{\pm 1.10}$ | $92.99^{\uparrow 39.38}_{\pm 1.17}$ | $93.25^{\uparrow 32.90}_{\pm 0.97}$ |

yielding improvements in test accuracy shown in Fig. 5c. Meanwhile, in Tab. 3 we further provide *geometric mean scores* (GM, a metric often used for imbalanced dataset (Kubat et al., 1997; Kim et al., 2020)), which is defined by the geometric mean over class-wise sensitivity for evaluate the classification performance of models trained in MNAR. More metrics for evaluation (e.g., precision and recall) and the results of PRG built on other SSL learner can be found in Appendix D.2.

Our main competitors can be divided into three categories. (1) State-of-The-Art (SOTA) SSL methods such as ReMixMatch (Berthelot et al., 2020) and FixMatch (Sohn et al., 2020). As shown in Tabs. 1 and 2, these methods show poor performance under MNAR. Especially, our backbone FixMatch can't cope with MNAR at all, whereas with our method, the performance is significantly improved by more than 10% in most cases. (2) Imbalanced SSL methods: DARP (Kim et al., 2020) and Crest (Wei et al., 2021). These two SOTA methods addressing long-tailed distribution in SSL emphasize the bias removal in matched distribution (i.e., the unlabeled data is equally imbalanced as the labeled data), showing very limited capacity in handling MNAR. (3) SSL solutions devised for the MNAR scenarios: CADR (Hu et al., 2022). Our method outperforms CADR under its proposed protocol across the board, demonstrating PRG is more effective for bias removal on label imputation than it. With extremely few labels, the class-aware propensity estimation in CADR is not reliable whereas our method still works well, yielding a performance gap of up to 14.41%.

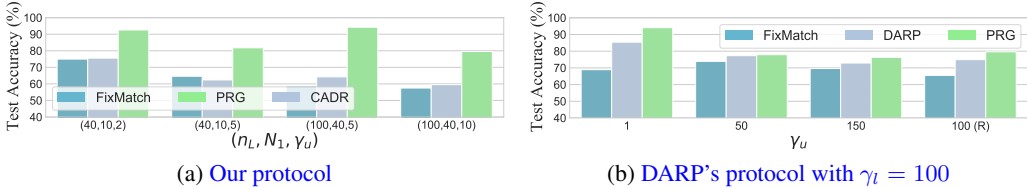

(a) Our protocol

(b) DARP's protocol with $\gamma_l = 100$

Figure 6: Results on CIFAR-10 under two protocols. The imbalanced distributions of labeled and unlabeled data are in reverse order of each other in (a) and the case of $\gamma_u$ marked with *"R"* in (b).

**More MNAR Settings.** More MNAR scenarios are considered for evaluation. In our protocol, we alter $N_1$ and $\gamma_u$ to mimic the case where the distributions of the labeled and unlabeled data are imbalanced and mismatched, *i.e.*, the two distributions are different. Likewise, DARP's protocol produces similar mismatched distributions. As shown in Fig. 6, PRG achieves promising results in all the comparisons with the baseline methods. Our method boosts the accuracy of the original FixMatch by up to 35.51% and 24.33% in our and DARP's protocols respectively. The activated class transitions make the model less prone to over-learning unexpected classes so that the negative effect of MNAR can be mitigated. Moreover, the results of balanced labeled data with imbalanced unlabeled data and more application scenarios (*e.g.*, tabular data) can be found in Appendix D.2.

## 4.2 CONVENTIONAL SSL SETTINGS AND ABLATION STUDIES ON HYPER-PARAMETERS

As shown in Tab. 4, our method still works well on balanced datasets. The class-level guidance offered by our method is also valid in the conventional setting while maintaining the vitality of class transition, even though there is not too much need to remove bias on label imputation. Hereafter, we investigate the effect of the class invariance coefficient $\alpha$ and the tracked batch number $N_B$ on PRG, which is shown in Fig. 7. Choosing an appropriate $\alpha$ to control the degree of class invariance in

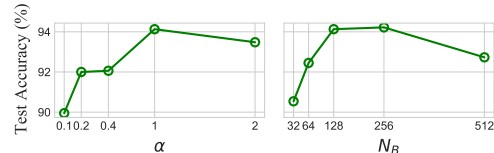

Figure 7: Ablation studies with $\alpha$ and $N_B$ on CIFAR-10 under CADR's protocol with $\gamma = 20$.

pseudo-rectifying is important for PRG, which ensures stability of supervision information and training. Meanwhile, we note that too small $N_B$ is not sufficient to estimate the underlying distribution of class transitions, where $N_B = 128$ is a sensible choice for both memory overhead and performance.

Table 4: Mean accuracy (%) in the conventional setting with various $n_L$. Results of baselines are reported in CADR (Hu et al., 2022) while results of $^*$ are based on our reimplementation.

| Method | CIFAR-10 | | | CIFAR-100 | | | mini-ImageNet |
|---|---|---|---|---|---|---|---|
| | $n_L = 40$ | 250 | 4000 | 400 | 2500 | 10000 | 1000 |
| FixMatch | $88.61_{\pm3.35}$ | $\mathbf{94.93}_{\pm0.33}$ | $95.69_{\pm0.15}$ | $50.05_{\pm3.01}$ | $\mathbf{71.36}_{\pm0.24}$ | $76.82_{\pm0.11}$ | $39.03_{\pm0.66}{}^*$ |
| + CADR | $94.41^{\uparrow5.80}$ | $94.35^{\downarrow0.58}$ | $95.59^{\downarrow0.10}$ | $\mathbf{52.90}^{\uparrow2.85}$ | $70.61^{\downarrow0.75}$ | $76.93^{\uparrow0.11}$ | - |
| + PRG (Ours) | $\mathbf{94.44}^{\uparrow5.83}_{\pm0.16}$ | $94.42^{\downarrow0.51}_{\pm0.06}$ | $95.38^{\downarrow0.31}_{\pm0.10}$ | $52.45^{\uparrow2.40}_{\pm3.75}$ | $70.12^{\downarrow1.24}_{\pm0.21}$ | $76.49^{\downarrow0.33}_{\pm0.42}$ | $47.34^{\uparrow8.31}_{\pm1.60}$ |
| + PRG$^{\text{Last}}$ (Ours) | $93.00^{\uparrow4.39}_{\pm0.79}$ | $94.43^{\downarrow0.50}_{\pm0.33}$ | $\mathbf{95.75}^{\uparrow0.06}_{\pm0.11}$ | $48.81^{\downarrow1.24}_{\pm0.15}$ | $70.01^{\downarrow1.35}_{\pm0.02}$ | $\mathbf{77.12}^{\uparrow0.30}_{\pm0.13}$ | $\mathbf{48.23}_{(+9.20)}$ |

## 5 CONCLUSION

This paper can be concluded as proposing a effective SSL framework called class transition based Pseudo-Rectifying Guidance (PRG) to address SSL in the MNAR scenarios. Firstly, we argue that the history of class transition caused by pseudo-rectifying can be utilized to offer informative guidance for future label assignment. Thus, we model the class transition as a Markov random walk along the nodes of the graph constructed on the class tracking matrix. Finally, we propose to utilize the class prediction information at current epoch (an alternative strategy is to combine the class prediction at the last epoch) to guide the class transition for pseudo-rectifying so that the bias of label imputation can be alleviated. Given that our method achieves considerable performance gains in various MNAR settings, we believe PRG can be used for robust semi-supervised learning in broader scenarios.

**Reproducibility Statement.** For reproducibility, please refer to the method described in Sec. 3 and the algorithmic presentation shown in Sec. A. The implementation details (including backbone, hyper-parameters, traning details, *etc*) can be found in Sec. 4 and Sec. C. Moreover, the checkpoints and evaluation code are available at the anonymous link `https://anonymous.4open.science/r/PRG4SSL-MNAR-8DE2`. We **promise** to release the all source code if the paper is accepted.

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

## APPENDIX

## A ALGORITHM

Pseudo-code of PRG is presented in Algorithm 1 while that of PRG$^{\text{Last}}$ is presented in Algorithm 2.

---

**Algorithm 1:** PRG: Pseudo-Rectifying Guidance

---

**Input:** class tracking matrices $\mathcal{C} = \{\mathbf{C}^{(i)}; i \in (1, ..., N_B)\}$, labeled training dataset $D_L$, unlabeled training dataset $D_U$, model $\theta$, label bank $\{l^{(i)}; i \in (1, ..., n_T - n_L)\}$

1 **for** $n = 1$ **to** MaxIteration **do**
2      From $D_L$, draw a mini-batch $\mathcal{B}_L = \{(x_L^{(b)}, y_L^{(b)}); b \in (1, ..., B)\}$
3      From $D_U$, draw a mini-batch $\mathcal{B}_U = \{(x_U^{(b)}); b \in (1, ..., B_U)\}$
4      $\mathbf{H} = \text{RowWiseNormalize}(\text{Average}(\mathcal{C}))$      // Construct transition matrix
5      $H'_{ij} = \frac{\sum_{d=1}^{k} L_d}{\sum_{d=1}^{k} \sum_{d'=1}^{k} C_{dd'}} \times \frac{\sum_{d=1}^{k} C_{id}}{L_j} \times H_{ij}$      // Rescale $\mathbf{H}$ at class/batch-level
6      **for** $b = 1$ **to** $B_U$ **do**
7          $p^{(b)} = f_\theta(x_U^{(b)})$      // Compute model prediction
8          $\text{idx} = \text{Index}(x_U^{(b)})$      // Obtain the index of $x_U^{(b)}$ in $D_U$
9          $\hat{p}^{(b)} = \arg\max(p^{(b)})$      // Compute class prediction
10         **if** $l^{(\text{idx})} \neq \hat{p}^{(b)}$ **then**
11             $C_{l^{(\text{idx})}\hat{p}^{(b)}}^{(n)} = C_{l^{(\text{idx})}\hat{p}^{(b)}}^{(n)} + 1$      // Perform class transition tracking
12             $l^{(\text{idx})} = \hat{p}^{(b)}$
13         **end**
14         $\tilde{p}^{(b)} = \text{Normalize}(H'_{\hat{p}^{(b)}} \circ p^{(b)})$      // Perform pseudo-rectifying guidance
15      **end**
16      $\mathcal{L}_L, \mathcal{L}_U = \text{FixMatch}\left(\mathcal{B}_L, \mathcal{B}_U, \{\tilde{p}^{(b)}; b \in (1, ..., B_U)\}\right)$      // Run FixMatch
17      $\theta = \text{SGD}(\mathcal{L}_L + \mathcal{L}_U, \theta)$      // Update model parameters $\theta$
18 **end**

---

**Algorithm 2:** PRG$^{\text{Last}}$: Pseudo-Rectifying Guidance Using Class Predictions of the Last Epoch

---

**Input:** class tracking matrices $\mathcal{C} = \{\mathbf{C}^{(i)}; i \in (1, ..., N_B)\}$, labeled training dataset $D_L$, unlabeled training dataset $D_U$, model $\theta$, label bank $\{l^{(i)}; i \in (1, ..., n_T - n_L)\}$

1 **for** $n = 1$ **to** MaxIteration **do**
2      From $D_L$, draw a mini-batch $\mathcal{B}_L = \{(x_L^{(b)}, y_L^{(b)}); b \in (1, ..., B)\}$
3      From $D_U$, draw a mini-batch $\mathcal{B}_U = \{(x_U^{(b)}); b \in (1, ..., B_U)\}$
4      $\mathbf{H} = \text{RowWiseNormalize}(\text{Average}(\mathcal{C}))$      // Construct transition matrix
5      $H'_{ij} = \frac{\sum_{d=1}^{k} L_d}{\sum_{d=1}^{k} \sum_{d'=1}^{k} C_{dd'}} \times \frac{\sum_{d=1}^{k} C_{id}}{L_j} \times H_{ij}$      // Rescale $\mathbf{H}$ at class/batch-level
6      **for** $b = 1$ **to** $B_U$ **do**
7          $p^{(b)} = f_\theta(x_U^{(b)})$      // Compute model prediction
8          $\text{idx} = \text{Index}(x_U^{(b)})$      // Obtain the index of $x_U^{(b)}$ in $D_U$
9          $\tilde{p}^{(b)} = \text{Normalize}(H'_{l^{(\text{idx})}} \circ p^{(b)})$      // Perform pseudo-rectifying guidance
10         $\hat{p}^{(b)} = \arg\max(p^{(b)})$      // Compute class prediction
11         **if** $l^{(\text{idx})} \neq \hat{p}^{(b)}$ **then**
12             $C_{l^{(\text{idx})}\hat{p}^{(b)}}^{(n)} = C_{l^{(\text{idx})}\hat{p}^{(b)}}^{(n)} + 1$      // Perform class transition tracking
13             $l^{(\text{idx})} = \hat{p}^{(b)}$
14         **end**
15      **end**
16      $\mathcal{L}_L, \mathcal{L}_U = \text{FixMatch}\left(\mathcal{B}_L, \mathcal{B}_U, \{\tilde{p}^{(b)}; b \in (1, ..., B_U)\}\right)$      // Run FixMatch
17      $\theta = \text{SGD}(\mathcal{L}_L + \mathcal{L}_U, \theta)$      // Update model parameters $\theta$
18 **end**

---

Table 5: The ablation studies on re-weighting by Eq. (6). We report the results (accuracy (%) / GM) on CIFAR-10 under CADR's protocol.

| Method | $\gamma = 20$ | $\gamma = 50$ | $\gamma = 100$ |
|---|---|---|---|
| PRG wo. Eq. (6) | 88.97 / 87.37 | 91.73 / 91.28 | 92.72 / 92.55 |
| PRG | 94.04 / 93.53 | 94.09 / 93.70 | 94.28 / 93.94 |

Table 6: The ablation studies on step $k$. We report accuracy (%) and GM on CIFAR-10 under CADR's protocol with $\gamma = 20$.

| $k$ | 1 (default PRG) | 2 | 5 | 10 |
|---|---|---|---|---|
| Accuracy | 94.04 | 91.33 | 87.76 | 82.60 |
| GM | 93.53 | 90.79 | 85.80 | 80.24 |

# B    DISCUSSION ON **H**-BASED PSEUDO-RECTIFYING GUIDANCE

In this section, we give insights into re-weighting scheme of $\mathbf{H}$ in Eq. (6) based on the following theoretical justification. Overall, we give an explanation from the perspective of gradient. Our re-weighting scheme potentially scale the gradient magnitude on the learning of the unlabeled data to mitigate adverse effects of biased labeled data, and suppresses the gradient magnitude when lass transition is overheated. Letting $p$ be the naive soft label vector, by Eq. (6), we re-weight $\mathbf{H}$ by $H'_{ij} = \frac{\sum_{d=1}^{k} L_d}{\sum_{d=1}^{k} \sum_{d'=1}^{k} C_{dd'}} \times \frac{\sum_{d=1}^{k} C_{id}}{L_j} \times H_{ij}$ and obtain the rescaled pseudo-label vector $\tilde{p} = \text{Normalize}(\mathbf{H}' \circ p)$. Hence, the cross-entropy between prediction $p$ and $\tilde{p}$ can be formalized as

$$\mathcal{L}_U = -\sum_{c}^{k} \tilde{p} \log p_c = -\sum_{c}^{k} \left( \frac{\frac{\sum_{d=1}^{k} L_d}{\sum_{d=1}^{k} \sum_{d'=1}^{k} C_{dd'}} \times \frac{\sum_{d=1}^{k} C_{\hat{p}d}}{L_c} \times H_{\hat{p}c} \times p_c}{\mathcal{Z}} \right) \log p_c$$

$$= -\frac{\sum_{d=1}^{k} C_{\hat{p}d}}{\mathcal{Z} \sum_{d=1}^{k} \sum_{d'=1}^{k} C_{dd'}} \sum_{c}^{k} \left( \frac{H_{\hat{p}c} \times p_c}{\mathcal{Z} \frac{L_c}{\sum_{d=1}^{k} L_d}} \right) \log p_c, \quad (8)$$

where $\mathcal{Z}$ is the normalize factor. $\frac{L_c}{\sum_{d=1}^{k} L_d}$ can be regarded as the ratio of pseudo-labels belonging to class $c$ to all labels and $\frac{\sum_{d=1}^{k} C_{\hat{p}d}}{\sum_{d=1}^{k} \sum_{d'=1}^{k} C_{dd'}}$ can be regarded as the ratio of class transitions derived from class $\hat{p}$ to population transitions. Denoting the logit outputted from the model as $o$ (implying $p = \text{Softmax}(o)$), with no gradient on pseudo-label $\tilde{p}$, we obtain $\frac{\partial \mathcal{L}_U}{\partial o_c} = -\sum_{c}^{k} \frac{\tilde{p}_c}{p_c} \frac{\partial p_c}{\partial o_c}$, i.e.,

$$\frac{\partial \mathcal{L}_U}{\partial o_c} = -(\tilde{p}_c - \tilde{p}_c p_c - \sum_{i \neq c}^{k} \tilde{p}_i p_c) = \sum_{i}^{k} \tilde{p}_i p_c - \tilde{p}_c = \frac{\sum_{d=1}^{k} C_{\hat{p}d}}{\mathcal{Z} \sum_{d=1}^{k} \sum_{d'=1}^{k} C_{dd'}} \left( 1 - \frac{H_{\hat{p}c}}{\mathcal{Z} \frac{L_c}{\sum_{d=1}^{k} L_d}} \right) p_c. \quad (9)$$

The larger the difference between $H_{\hat{p}c}$ and $\frac{L_c}{\sum_{d=1}^{k} L_d}$, the larger the gradient; and the smaller the difference between $H_{\hat{p}c}$ and $\frac{L_c}{\sum_{d=1}^{k} L_d}$, the smaller the gradient ($\frac{\partial \mathcal{L}_U}{\partial o_c} = 0$ when $\frac{H_{\hat{p}c}}{\mathcal{Z} \frac{L_c}{\sum_{d=1}^{k} L_d}} = 1$). This means that we intend to provide unbiased guidance (because this is derived from the unlabeled data) for the learning of unlabeled samples from the class level, so as to resist the influence of biased labeled samples. Meanwhile, if there are too many class transitions occur on the whole, $\frac{\sum_{d=1}^{k} C_{\hat{p}d}}{\sum_{d=1}^{k} \sum_{d'=1}^{k} C_{dd'}}$ will decrease, which results in decreasing of $\mathcal{L}_U$, i.e., suppress the trend of class transition overheating. To demonstrate the effectiveness of our re-weighting scheme on $\mathbf{H}$, we conduct ablation experiments on it. As shown in Tab. 5, the re-weighting scheme can effectively boost the performance of PRG in MNAR because it controls the intensity of class transition together. Additionally, for the utilization of $\mathbf{H}'$ in Eq. (7), we consider taking $k$ steps, i.e., multiply by $\mathbf{H}'^k$ instead of $\mathbf{H}'$ to uncover more complex patterns of misclassification than simple pairwise class relations. However, as shown in Tab. 6, we can observe that the performance is inversely proportional to $k$. The advantage of PRG is that $\mathbf{H}'$ is updated in each iteration, which means that the value of $\mathbf{H}'$ is dynamic. As the model learns new knowledge, the past $\mathbf{H}'$ may not be suitable for the pseudo-rectifying process anymore. If $\mathbf{H}'^k$ is used, this means that we are using the same $\mathbf{H}'$ multiple times for a given sample, which wastes the advantage of dynamic $\mathbf{H}'$. $\mathbf{H}'^k$ using a suitable $k$ or a dynamic selection of $k$ might yield better performance, but it is difficult for us to determine the value of $k$. Therefore, the PRG is designed for simplicity and exhibits superior the performance.

## C  IMPLEMENTATION DETAILS

In this section, we show the complete hyper-parameters in Tab. 7. As mentioned in Sec. 4, our method is implemented as a plugin to FixMatch (Sohn et al., 2020). Thus, we keep the original hyper-parameters in FixMatch and alert additional hyper-parameters in our method. Note that FixMatch sets different values of weight decay $w$ for CIFAR-10 and CIFAR-100, which are 0.0005 and 0.001 respectively. For simplicity, we set $w = 0.0005$ for all experiments in our work. Additionally, the models in this paper are trained on GeForce RTX 3090/2080 Ti and Tesla V100. We observe that since no additional network components are introduced, the average running time of single iteration hardly increased, which means our method does not introduce excessive computational overhead.

Table 7: Complete list of hyper-parameters of PRG plugged in FixMatch Sohn et al. (2020). $N_B$ and $\alpha$ are additional hyper-parameters in our method whereas other hyper-parameters follow FixMatch. Note that the unlabeled data batch size can be calculated by $B_U = \mu B$.

| Hyper-parameter | Description | CIFAR-10 | CIFAR-100 | mini-ImageNet |
|---|---|---|---|---|
| $\mu$ | The ratio of unlabeled data to labeled data in a mini-batch | | 7 | |
| $B$ | Batch size for labeled data and class transition tracking | | 64 | |
| $B_U$ | Batch size for unlabeled data | | 448 | |
| $\lambda_U$ | Unlabeled loss weight | | 1 | |
| $\tau$ | Confidence threshold | | 0.95 | |
| $lr$ | Start learning rate | | 0.03 | |
| $\beta$ | Momentum | | 0.9 | |
| $w$ | Weight decay | | 0.0005 | |
| $N_B$ | Tracked batch number | | 128 | |
| $\alpha$ | Class invariance coefficient | | 1 | |

## D  ADDITIONAL EXPERIMENTAL RESULTS

### D.1  USING DISTRIBUTION ALIGNMENT IN MNAR

As discussed in Sec. 3.2, *distribution alignment* (DA) aims to perform strong regularization on pseudo-labels by aligning the class distribution of predictions on unlabeled data to that of labeled data. DA boosts the performance of SSL models tangibly (Berthelot et al., 2020; Gong et al., 2021; Li et al., 2021; Sohn et al., 2020). However, DA works on a strong assumption that the distribution of unlabeled data matches that of labeled data. In MNAR, this assumption does not hold obviously. Thus, these methods combining DA fail to address SSL in MNAR, eventually yielding abysmal performance. As shown in Tab. 8, rather than improving performance, integrating DA into SSL models is counterproductive, *e.g.*, original FixMatch outperforms FixMatch with DA by up to 28.68% on CIFAR-10. DA leads to a substantial deterioration of the model performance in the MNAR scenarios due to the large gap between the labeled data utilized and the unlabeled data distribution. Conversely, our method is not restricted by the mismatched distributions and achieves superior performance across the board, because PRG helps the model to better handle MNAR scenarios without any prior information (distribution prior estimated from labeled data is used in DA).

Table 8: Mean accuracy (%) in MNAR under our protocol compared with more baseline methods. **DA** indicates distribution alignment technique proposed in Berthelot et al. (2020). CoMatch (Li et al., 2021) is a recently-proposed SSL method integrating contrastive learning and graph-based methods. Note that CoMatch also combines DA to improve the quality of pseudo-labels for better performance in the conventional SSL setting.

| Method | CIFAR-10 ($n_L = 40$) | | CIFAR-10 ($n_L = 250$) | | CIFAR-100 ($n_L = 2500$) | | mini-ImageNet ($n_L = 1000$) | |
|---|---|---|---|---|---|---|---|---|
| | $N_1 = 10$ | 20 | 100 | 200 | 100 | 200 | 40 | 80 |
| CoMatch | 60.27 | 39.48 | 57.87 | 26.77 | 48.02 | 30.08 | 30.24 | 21.47 |
| FixMatch | 85.72 | 76.53 | 69.76 | 46.53 | 61.31 | 41.38 | 36.20 | 28.33 |
| + DA | $71.23^{\downarrow 14.49}$ | $47.85^{\downarrow 28.68}$ | $61.8^{\downarrow 7.96}$ | $27.61^{\downarrow 18.92}$ | $50.94^{\downarrow 10.37}$ | $31.82^{\downarrow 9.56}$ | $33.87^{\downarrow 2.33}$ | $23.53^{\downarrow 4.78}$ |
| + PRG (Ours) | $91.87^{\uparrow 6.15}$ | $77.44^{\uparrow 0.91}$ | $93.93^{\uparrow 24.17}$ | $67.86^{\uparrow 21.33}$ | $61.49^{\uparrow 0.18}$ | $49.84^{\uparrow 8.46}$ | $39.99^{\uparrow 3.79}$ | $35.39^{\uparrow 7.069}$ |
| + PRG$^{\text{Last}}$ (Ours) | $85.66^{\downarrow 0.06}$ | $77.85^{\uparrow 1.32}$ | $92.8^{\uparrow 23.04}$ | $64.0^{\uparrow 17.47}$ | $60.41^{\downarrow 0.90}$ | $43.8^{\uparrow 2.42}$ | $39.84^{\uparrow 3.64}$ | $33.10^{\uparrow 4.77}$ |

Table 9: The comparisons of Class-wise precision and recall on CIFAR-10 in the training process under CADR's protocol with $\gamma = 50$.

| Method | Class Index | 30000 Iterations | | 90000 Iterations | | 150000 Iterations | |
|---|---|---|---|---|---|---|---|
| | | Precision | Recall | Precision | Recall | Precision | Recall |
| FixMatch | 1 | 45.21 | 95.22 | 46.89 | 96.72 | 47.93 | 97.80 |
| | 2 | 49.12 | 99.01 | 49.59 | 99.27 | 50.27 | 98.72 |
| | 3 | 38.49 | 88.73 | 39.74 | 88.43 | 70.26 | 89.47 |
| | 4 | 75.02 | 68.13 | 75.63 | 72.19 | 82.04 | 75.93 |
| | 5 | 86.14 | 88.43 | 86.88 | 90.21 | 88.42 | 94.38 |
| | 6 | 89.45 | 62.93 | 91.03 | 64.4 | 89.31 | 75.98 |
| | 7 | 86.47 | 90.03 | 90.23 | 8.89 | 91.37 | 94.80 |
| | 8 | 89.09 | 75.94 | 90.48 | 75.21 | 95.32 | 75.37 |
| | 9 | 99.02 | 0.00 | 97.95 | 1.00 | 97.21 | 2.00 |
| | 10 | 0.00 | 0.00 | 99.60 | 0.33 | 98.60 | 0.67 |
| + PRG (Ours) | 1 | 70.52 | 93.52 | 87.34 | 95.50 | 88.25 | 95.34 |
| | 2 | 82.53 | 98.21 | 96.03 | 98.32 | 96.78 | 98.56 |
| | 3 | 73.52 | 76.54 | 90.92 | 89.85 | 92.37 | 90.57 |
| | 4 | 70.21 | 73.77 | 85.36 | 80.51 | 87.89 | 81.37 |
| | 5 | 79.03 | 86.57 | 90.31 | 96.31 | 92.74 | 96.19 |
| | 6 | 74.55 | 61.03 | 90.58 | 79.88 | 90.97 | 82.43 |
| | 7 | 89.12 | 91.40 | 93.09 | 97.02 | 93.79 | 98.03 |
| | 8 | 92.58 | 80.14 | 95.01 | 96.21 | 96.32 | 97.50 |
| | 9 | 96.31 | 76.50 | 95.22 | 92.12 | 95.63 | 93.55 |
| | 10 | 96.56 | 62.52 | 96.95 | 96.01 | 97.15 | 96.81 |

## D.2 MORE EVALUATIONS ON PRG

**More Metrics.** To comprehensively explore the improvement of PRG in MNAR, we report the difference in class-wise precision and recall with/without PRG. The experimental results are shown in Tab. 9. Compared to original FixMatch, we witness FixMatch with PRG achieves higer precision/recall by and large, especially on rare classes (*i.e.*, class with larger index), which demonstrates that the bias removal capability of PRG effectively mitigates the effect of MNAR on the model. We also observe that both PRG and FixMatch achieve high precision as well as recall on popular classes and high precision but low recall on rare classes (especially FixMatch) in the early training period. The improvement of recall by PRG is due to the activated class transitions, which gives the model

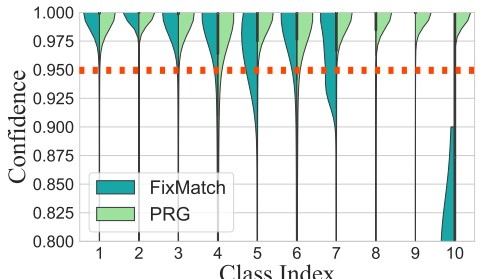

Figure 8: Violin plot of confidence scores on the unlabeled data under CADR's protocol with $\gamma = 100$. The confidence threshold $\tau = 0.95$ in FixMatch is marked out in red.

a certain probability to assign pseudo-labels to rare classes. In addition, as shown in Fig. 8, PRG exhibits superior bias removal for confidence of pseudo-labels in MNAR, whereas FixMatch filters out too many labeled rare class samples with confidence lower than $\tau$, *e.g.*, class 8, 9 and 10, resulting in the waste of unlabeled data.

**More MNAR Scenarios** We also provide more experiments on the setting of balanced labeled data with imbalanced unlabeled data, which is summarized in Tab. 10. For specific, we set $n_L = 40$ with balanced distribution and set $\gamma_u = 50, 100, 200$ for imbalanced unlabeled data, *i.e.*, the class-wise number of unlabeled data $M_i = M_1 \times \gamma_u^{-\frac{k-i}{k-1}}$, where $M_1 = 5000$ in CIFAR-10. As shown in Tab. 10, PRG outperforms all baseline methods by a large margin (it is worth noting that the performance of CADR is even weaker than original FixMatch), proving the robustness of PRG in this MNAR scenario due to the unbiased guidance derived from the class transition history.

Table 10: Accuracy (%) on CIFAR-10 with $n_L = 40$ and various $\gamma_u$. The labeled data is balanced and the unlabeled data is imbalanced.

| Method | $\gamma_u = 20$ | $\gamma_u = 50$ | $\gamma_u = 100$ |
|---|---|---|---|
| CoMatch | 52.73 | 46.20 | 38.85 |
| FixMatch | 57.54 | 54.82 | 50.67 |
| + DA | 54.08$^{\downarrow 3.46}$ | 46.71$^{\downarrow 8.11}$ | 41.37$^{\downarrow 9.30}$ |
| + CADR | 49.38$^{\downarrow 8.16}$ | 45.27$^{\downarrow 9.55}$ | 42.30$^{\downarrow 8.37}$ |
| + PRG (Ours) | 62.43$^{\uparrow 4.90}$ | 62.44$^{\uparrow 7.62}$ | 58.23$^{\uparrow 7.56}$ |

Table 11: Accuracy (%) on tabular MNIST. $\gamma$ is varied for CADR's protocol whereas $n_L$ and $N_1$ are varied for our protocol. `Mean` $\pm$ `Std`. are computed over 50 runs.

| Method | $\gamma = 20$ | 50 | 100 | $n_L, N_1 = 40, 10$ | 40, 20 | 250, 100 | 250, 200 |
|---|---|---|---|---|---|---|---|
| VIME | 63.38±4.42 | 63.75±6.10 | 64.80±2.76 | 50.13±7.56 | 30.73±8.69 | 60.58±2.68 | 21.44±0.58 |
| + PRG (Ours) | 59.41±14.45 | 65.92±13.9 | 66.60±12.58 | 49.28±11.09 | 34.08±16.05 | 66.14±11.88 | 24.51±9.56 |

**More SSL Learners.** Moreover, to further evaluate PRG's performance, we consider building PRG on the top of more SSL frameworks. Thus, we conduct experiments on CIFAR-10 under CADR's protocol with UPS (Rizve et al., 2021) combining PRG. UPS is a recently-proposed uncertainty-aware pseudo-label selection framework for SSL, which is the SOTA method among pseudo-labeling based methods. We keep all training settings the same as the original UPS. With $\gamma = 20$, UPS achieves an accuracy of **30.46%** whereas UPS with PRG achieves an accuracy of **32.22%**. We note that UPS performs poorly in the MNAR scenarios because it is a more pure pseudo-labeling approach that does not introduce consistency regularization to improve model performance. Also we observe that PRG improves UPS marginally, much less than FixMatch. This is understandable because the negative learning that UPS prides itself on can be potentially negatively affected by the probability distribution of pseudo-label being adjusted by PRG, *e.g.*, uncertainty being altered.

**More Data Type** In order to explore a broader application scenario of PRG, we apply it to tabular data. We conduct further experiments on tabular MNIST (interpreting MNIST as tabular data with 784 features) by plugging PRG into VIME (Yoon et al., 2020). VIME is a prevailing self- and semi-supervised learning frameworks for tabular data with pretext task of estimating mask vectors from corrupted tabular data. We implement PRG above the semi-supervised learning component of VIME. PRG provide pseudo-rectifying guidance to rescale the pseudo-labels for the original unlabeled sample in VIME. Specially, we replace the consistency loss used in VIME (*i.e.*, mean squared error in Eq. (9) in Yoon et al. (2020)) with standard cross-entropy loss to makes PRG applicable to VIME. We use two protocols to show the performance advantage of PRG, including CADR's protocol and our protocol. As shown in Tab. 11, except for CADR's protocol with $\gamma = 20$ and our protocol with $n_L = 40, N_1 = 10$ (it is worth noting that our upper limits of performance greatly exceed that of VIME), PRG outperforms original VIME by a large margin in the most of settings. The reason is that the scheme of class-transition-based pseudo-rectifying guidance is high-level and general (not limited to image data), which ultimately yields robust effect on the MNAR problem with tabular data.

