# OpenReview forum: "Towards Semi-Supervised Learning with Non-Random Missing Labels"
_ICLR.cc/2023/Conference — Submitted to ICLR 2023_

### Official Review · Reviewer_buPA · 2022-10-26

**Confidence:** 3
**Correctness:** 3
**Technical Novelty And Significance:** 2
**Empirical Novelty And Significance:** 3
**Recommendation:** 5

**Clarity, Quality, Novelty And Reproducibility:**

To the best of my knowledge the ideas in the paper are novel, and I have no reason to believe the results are not reproducible. The presentation of the paper leaves something to be desired. The overall motivation is not exposed clearly, as discussed previously, and the text has a number of typos and some rather unclear parts. For instance, I failed to understand the sentences below.
- "The propensity of rectifying the labels belonging to the same class to different classes is different".
- "PRG recalls those “old face” classes once assigned to the unlabeled data".

**Strength And Weaknesses:**

### Strength
- The paper proposes a simple and seemingly effective method that improves previous approaches for semi-supervised learning with imbalanced and mismatched class distributions. That is a relevant contribution that, I believe, would be useful both in practical applications and future research.
- Empirically the proposed method works well, outperforming previous competing methods by significant margins.

### Weaknesses
- The motivation of the proposed methods is somewhat weak.
    - The definition of the transition matrix $\bf H$ is not very well motivated. It is not entire clear to me why Equation (6) is preferable to simply estimating $\bf H$ from the transitions observed in a batch of data. At least that would be a simple and telling baseline to compare against that, I believe, would add to the paper.
    - The authors say "the history of class transitions can point the way for bias removal on label imputation with an abnormal propensity on different classes caused by mismatched distributions in MNAR" but only hint as to why that is the case. We are told PRG works because it encourages the model to predict under-represented classes, but that does not necessarily require a history of transitions nor does it attenuate "over-learning the samples of the labeled popular classes". Empirically, PRG seems to work well, supporting the authors' claim to some extent, but in the current version of the paper the connection between the history of transitions and mismatched class distributions is rather loose. To be clear, I am not suggesting such a connection does not exist. I am just stating that it is not obvious (and thus potentially interesting) and the paper in its current version does not explain it very clearly.
    - The random walk motivation is interesting but feels under-explored. Why take a simple step in the random walk? If we took $k$ steps, i.e., multiply by $\bf H^k$ instead of $\bf H$ in Equation (7), we could uncover more complex patterns of misclassification than simple pairwise class relations.  For instance, if classes A and B are often misclassified as class C, but the model rarely classifies A as B and vice-versa, PRG would not capture this sort of relationship even though the transition matrix does.

- Given that the motivation of the paper is not very grounded, the experiments seem not thorough enough since they are limited to CIFAR and mini-imagenet. CIFAR is notably noisy, which my benefit PRG. It would be interesting to see if the history of transitions is a useful artefact for general types of data other than images. Even simple and easy to run tabular datasets would already contribute to the overall message of the paper.

- The paper is harder to follow than one would expect for such a simple and intuitive approach. Parts of the text are confusing and some of the notation is unnecessary or not properly introduced. I comment on some examples of this in the questions and minor issues below, and in the clarity section.

### Questions
- What exactly is $\xi$ computing? "Rectifying weight vector" is not a very precise description.
- In Equation (6), isn't $\sum_{d=1}^k L_d$ always equal to 1? So the first term on right-hand side of (6) is just the ratio of total number of predictions over the number of predictions that changed class (in a given batch), and is thus always larger than one?
- If $\bf H$ is a proper transition matrix, while do we need to renormalise $p$? Is it because of the $\xi$ term?
- In Figure 7, it is interesting that the performance drops for larger batch sizes. One would expect the estimate of the transition matrix to be improve, which in turn should translate into better or at least similar performance. Do the authors have an idea as why that is the case? Could it be that the learning rate, if kept fixed in the ablation study, was too large for a batch size of 512?

### Minor issues
- The authors use $\max(p)$ as a measure of the model's confidence. That is only one of the possible ways to quantify confidence and that choice should be mentioned more explicitly. Also, the authors imply that the model is more confident ($\max(p)$ is higher) when classifying classes that appear frequently. While this might be intuitive, I think this is an empirical observation that does not always hold for every data point and for arbitrary models, and in particular not for neural networks. If this is indeed an empirical observation, it should be made clear.
- In Figure 4, I cannot see the pattern the authors describe in "the lower right corner (corresponding to the labeled popular classes) of the heatmap is getting lighter and always lighter than the upper left corner." Maybe the Figure is not showing right or the matrices need to be renormalised to show the desired effect?
- Probably best to mention $C_{ii}=0$ already in Equation (5).
- The missing label indicator $m$ is only used once. Not sure it is worth introducing it.

**Summary Of The Paper:**

The paper tackles semi-supervised learning with a focus on the most challenging setting of when there is a mismatch between the distributions of labelled and unlabelled data. The proposed method, PRG, uses the history of class transitions to sway the pseudo-labelling mechanism towards under-represented classes. This works by exploiting pairwise similarities between classes: classes that are often confused must be similar (with respect to the predictions of the model) and thus, if the model predicts one of them, we should also consider the others as possible pseudo-labels with certain probability. The authors validate PRG on CIFAR and mini-Imagenet with promising results, outperforming previous similar methods.

**Summary Of The Review:**

The main idea of the paper is interesting and seems to work well in a relevant scenario. The methods are rather simple, but I see that as a potential strength, facilitating their application in future work. However, the exposition of the paper is not clear and the main ideas and motivation are under-explored or not conveyed plainly.

---

> ### Author Response · Authors · 2022-11-16
> **Response to Reviewer buPA (6/6)**
>
> > Minor issue 3) Probably best to mention $C\_{ii}=0$ already in Equation (5).
>
> Thank you for your helpful suggestion. We have mentioned $C\_{ii}=0$ in Eq. (5) in the new revision.
>
> > Minor issue 4) The missing label indicator $m$ is only used once. Not sure it is worth introducing it.
>
> In CADR [2] (first to introduce MNAR to SSL), $m$ is introduced to formally define the problem of SSL with non-random missing labels. In order to make the symbols consistent for the MNAR problem in the SSL community, we follow the usage of $m$ in [2], which makes the description of the problem clearer when writing.
>
> > Minor issue 5) I failed to understand the sentences below. (1) "The propensity of rectifying the labels belonging to the same class to different classes is different". (2) "PRG recalls those “old face” classes once assigned to the unlabeled data".
>
> We are sorry for our confusing statement and have updated it in the new revision:
> - (1) "**Intuitively, given $p$ (pseudo-label) with $\hat{p}=i$ (class prediction), the model prefers to rectify it to another class similar to class $i$ in class transition**, i.e., the preference of class transitions can also be regarded as the similarity between classes and the more similar two classes are, the more likely they are to be misclassified as each other's classes."
> - (2) "PRG recalls classes that are easily overlooked but appear in class transition history."
>
> ## References
>
> [1] Jinsung Yoon, Yao Zhang, James Jordon, and Mihaela van der Schaar. Vime: Extending the success of self-and semi-supervised learning to tabular domain. In Advances in Neural Information Processing Systems, 2020
>
> [2] Xinting Hu, Yulei Niu, Chunyan Miao, Xian-Sheng Hua, and Hanwang Zhang. On non-random missing labels in semi-supervised learning. In International Conference on Learning Representations, 2022.
>
> [3] Kihyuk Sohn, David Berthelot, Chun-Liang Li, Zizhao Zhang, Nicholas Carlini, Ekin D. Cubuk, Alex Kurakin, Han Zhang, and Colin Raffel. Fixmatch: Simplifying semi-supervised learning with consistency and confidence. In Advances in Neural Information Processing Systems, 2020.
>
> [4] Bowen Zhang, Yidong Wang, Wenxin Hou, Hao Wu, Jindong Wang, Manabu Okumura, and Takahiro Shinozaki. Flexmatch: Boosting semi-supervised learning with curriculum pseudo labeling. In Advances in Neural Information Processing Systems, 2021.
>
> ---
>
> Part (6/6)
>
> Thank you for reading this response. We would be happy to discuss any further questions!

---

> > ### Comment · Reviewer_buPA · 2022-11-27
> > **Thanks for the comprehensive rebuttal**
> >
> > I thank the authors for their reply, especially for the extra experimental results, which I think contribute to the paper. However, in spite of the good empirical results, I think the exposition of the paper still leaves a lot to be desired. Even after reading the new version of the paper and the authors' response a few times, I still fail to fully grasp the method and its motivation.
> >
> > I comment on a few issues below.
> >
> > - The new appendix B shows how the proposed method affects the gradients, but that is not a convincing theoretical justification for the the re-weighting scheme in $H$. We are told the method is justified because it induces a rescaling of the gradients similar to the rescaling in Equation (6). However, the motivation for such rescaling is empirical, not theoretical, and unfortunately poorly explained in the paper.
> >
> > - The experiments with $H^k$ are interesting and I thank the authors for that, but the rationale for why the method works poorly for $k>1$ is not convincing. For each $H$ we have a random walk which, if I understood correctly, is valid for a particular epoch. Taking multiple steps in such a random walk, is not necessarily a "waste". It is perfectly valid (and common), and as I mentioned in the original review $k>1$ could be important to capture complex similarity relations in the data (for a given $H$). It might not work because the data used in the experiments does not have this sort of similarity or because $H$ is poorly estimated or some other reason.
> >
> > - I still do not see the pattern discussed in the caption of Fig. 4. In all 5 graphs, transitions among rare classes are barely visible.

---

> > > ### Author Response · Authors · 2022-11-28
> > > **Response to Reviewer buPA - Round 2 (1/2)**
> > >
> > > ### Thank you for your further reply. Below is our response to your comments. We sincerely hope that our reply could address your concerns.
> > >
> > > ---
> > >
> > > > Comment 1) The new appendix B shows how the proposed method affects the gradients, but that is not a convincing theoretical justification for the the re-weighting scheme in $\mathbf{H}$. We are told the method is justified because it induces a rescaling of the gradients similar to the rescaling in Equation (6). However, the motivation for such rescaling is empirical, not theoretical, and unfortunately poorly explained in the paper.
> > >
> > > Thank you for your comments. As we mentioned in the paper and in the rebuttal, our motivation of re-weighting scheme on $\mathbf{H}$ can be concluded as: (1) We tend to suppress the trend of class transition overheating. (2) We tend to provid unbiased guidance with $\mathbf{H}$ derived from the unlabeled data at the class level, so as to resist the influence of biased labeled samples. Without this re-weighting scheme, pure PRG can not work well at the **class level** (because there is no participation of estimated **class distribution**, i.e., $\frac{L\_{c} }{\sum^{k}\_{d=1}L\_{d} }$), which is exactly what we are trying to do to fight against MNAR (after all, the trouble MNAR brings us is the **mismatch of class distribution**). The above discussion is according to the analysis of gradient in our response to Weakness 1): *"The larger the difference between $H\_{\hat{p}c}$ and $\frac{L\_{c} }{\sum^{k}\_{d=1}L\_{d} }$, the larger the gradient; Correspondingly, the smaller the difference between $H\_{\hat{p}c}$ and $\frac{L\_{c} }{\sum^{k}\_{d=1}L\_{d} }$, the smaller the gradient ($\frac{\partial \mathcal{L}\_{U}}{\partial o\_{c}}=0$ when $\frac{H\_{\hat{p}c}}{\mathcal{Z}\frac{L\_{c} }{\sum^{k}\_{d=1}L\_{d} }}=1$)."* This means we encourage the model to follow the guidance provided by $\mathbf{H}$ from the class-level (i.e., $\frac{H\_{\hat{p}c}}{\mathcal{Z}\frac{L\_{c}}{\sum^{k}\_{d=1}L\_{d}}}$ yields the effect of adjusting the class distribution of pseudo-labels), which enables $\mathbf{H}$ to play a better role in enhancing the model to preserve a certain probability to generate class transitions based on transition histroy (also corresponds to our initial motivation: providing pseudo-rectifying guidance at the **class level** to address MNAR). In view of the fact that we cannot revise the paper at the current rebuttal stage, we will add the above discussion to the future version.
> > >
> > > Following your suggestion, we have tried our best to make some justifications from the theoretical angle. However, we wish to clarify that we do not deny that the re-weighting scheme is more empirical than theoretical. At the same time, we notice that some methods in the current SSL community seem to be heuristic methods based on experimental observation and have achieved popularity and success, such as FixMatch, one of the most important SSL works in recent years. Similarly, the re-weighting scheme is also driven by methodology, and the ablation experiment we provided in the rebuttal (Tab. a) also proves its effectiveness. Thank you again for your comments. We will be committed to providing more theoretical perspectives and rethinking in the future to enable class transition technique to promote the development of SSL.
> > >
> > > ---
> > >
> > > Part (1/2)

---

> > > > ### Author Response · Authors · 2022-11-28
> > > > **Response to Reviewer buPA - Round 2 (2/2)**
> > > >
> > > > > Comment 2) The experiments $\dots$ It might not work because the data used in the experiments does not have this sort of similarity or because  is poorly estimated or some other reason.
> > > >
> > > > Thank you for your further comments. Below is our response.
> > > >
> > > > - First of all, we are sorry for the improper use of "waste". Our original idea is that when one transition can make the effect good enough, we can use $\mathbf{H}$ only once in each epoch to maximize the advantage of our dynamically updated $\mathbf{H}$, instead of focusing on one $\mathbf{H}$ for multiple uses. We would like to clarify our point that because the model is constantly updated ($\mathbf{H}$ is constantly updated and changing), the latest $\mathbf{H}$ is the most correct in probability. The larger $k$ is, the more uncertain/erroneous historical transitions are exploited, which could have a negative impact on the rectifying of pseudo-labels, i.e., the series of $\mathbf{H}$ over time estimated from unlabeled data is unstable. The premise of using the transition probability matrix to mine high-order and more complex inter-class potential relationships is that the transition probability matrix is reliable and stable, which is different from the scenario where we use $\mathbf{H}$. In other words, as the model is continuously updated, the latest estimate of $\mathbf{H}$ is relatively the most reliable. So we give up the high-order utilization of $\mathbf{H}$, and directly use the latest $\mathbf{H}$ (more likely to be accurate)  to adjust the pseudo-label.
> > > >
> > > > - At the same time, we have fully understood the inspiration you provided. We appreciate that you mentioned that $k>1$ in $\mathbf{H}^k$ could indeed explore more complex transition information across different classes. As in our previous rebuttal, we have actually provided the new experimental results. We can observe that using a large $k$ would lead to performance degeneration in Tab. b of our rebuttal. However, for the issue of utilizing $\mathbf{H}^k$ ($k>1$) as suggested, we wish to clarify this way actually computes the high-order transition way which is fundamentally different from our original goal. Taking the transition way of "dog" to "cat" in $\mathbf{H}^2$ as an example,  it actually computes how many an intermediate class x—with a role as "bridge"—to link the transition of "dog to x" and "x to cat" in $\mathbf{H}$. In this sense, the way of $\mathbf{H}^2$ is actually equivalent to a two-order transitive relation on a given set of classes. This is similar for other $\mathbf{H}^k$ ($k>1$), for example, $\mathbf{H}^5$ indicates the five-order transitive relation. We believe this complex transition, especially when $k$ is large, could overwhelm the class transition in SSL. For example, frequent transition only indicates these two classes are similar to each other. In this sense, using a typical $k>1$ to model $\mathbf{H}^k$, we might ignore the plainest direct transition way in $\mathbf{H}$, which is an efficient and effective way to model transition. This also corresponds to the result in Tab. b, i.e., a single transition is good enough. Finally, we very much agree that your suggestion is very enlightening, which will help us to further explore the in-depth mechanism of class transition in the future version.
> > > >
> > > > > Comment 3) I still do not see the pattern discussed in the caption of Fig. 4. In all 5 graphs, transitions among rare classes are barely visible.
> > > >
> > > > We are sorry that we neglected to update the description in the caption of Fig. 4. **We would like to clarify that we have updated the analysis of the experimental phenomenon for Fig. 4 in the body of Sec. 3.2 in revision (highlighted in blue), which also can be found in our response to Minor issue 2)**. Our original idea is to point out that the transitions involving rare classes gradually decreas, i.e., the transitions to or from rare classes, rather than the transitions "*between*" rare classes. In the body of the new version, we have changed this description to: "*The lower left corner and upper right corner of the heatmap (i.e., the class transitions between the popular classes and rare classes) is getting lighter and always lighter than the upper left corner (i.e., the class transitions among the popular classes), which means the model is increasingly reluctant to transfer the class prediction to the rare classes during the pseudo-rectifying  process.*"
> > > >
> > > > Since we cannot revise the paper at the current rebuttal stage, we will revise this caption in the future version as follows: **Class transitions occur intensively between the popular classes, and class transitions between the popular classes and rare classes gradually disappear (e.g., from "airplane" to "ship", or vice versa).**
> > > >
> > > > ---
> > > >
> > > > Part (2/2)
> > > >
> > > > Thank you for reading our response. We sincerely hope that your concerns can be addressed.

---

> ### Author Response · Authors · 2022-11-16
> **Response to Reviewer buPA (5/6)**
>
> > Question 4) In Figure 7, it is interesting that the performance drops for larger batch sizes. One would expect the estimate of the transition matrix to be improve, which in turn should translate into better or at least similar performance. Do the authors have an idea as why that is the case? Could it be that the learning rate, if kept fixed in the ablation study, was too large for a batch size of 512?
>
> Fisrt of all, we would like to clarify the definition of $N\_{B}$. **$N\_{B}$ is not the batch size for class transition tracking, but the tracked batch number**, i.e., the larger the $N\_{B}$, the more batches we tracked.
>
> The cause of the performance drops phenomenon is similar to our response to your Weakness 2). When the batch size used for class-transition tracking is too large, the update of $\mathbf{H}$ becomes sluggish accordingly, and the outdated $\mathbf{H}$ may no longer fit into the model update. We choose the appropriate $N\_{B}$ to be able to update $\mathbf{H}$ as quickly as possible while still being able to to estimate the underlying distribution of class transitions. Moreover, to address your concern, we provide additional  experiments on varying learning rate for the ablation study in Fig. 7. As shown in Tab. e, the performance of $N\_{B}=512$ is always lower than the original PRG (i.e., $N\_{B}$=128), which confirms our statement.
>
> Table e: Results on CIFAR-10 under CADR's protocol ($\gamma=20$) with $N\_{B}=512$ and various learning rates. The default PRG achieve an accuracy of 94.04%.
>
> |Learning rate|$0.3$ |$0.1$ | $0.05$ |  $0.01$
> | :-----:| :-----:| :----: | :----: | :----: |
> |Accuracy (%)| 92.33| 89.66| 82.91| 80.92 |
>
> ---
> ### Next, we give our response to your minor issues:
>
> > Minor issue 1) The authors use $\max(p)$ as a measure of the model's confidence. That is only one of the possible ways to quantify confidence and that choice should be mentioned more explicitly. Also, the authors imply that the model is more confident ($\max(p)$ is higher) when classifying classes that appear frequently. While this might be intuitive, I think this is an empirical observation that does not always hold for every data point and for arbitrary models, and in particular not for neural networks. If this is indeed an empirical observation, it should be made clear.
>
> We do admit that $\max(p)$ is only one of the possible ways to quantify confidence, which is widely used in previous SSL works, such as CADR [2], FixMatch [3], FlexMatch [4], etc. Thus, we have mentioned it more explicitly in Sec.3 in the new revision,
>
> For the statement that the model is more confident ($\max(p)$ is higher) when classifying classes that appear frequently, we do admit that this is an empirical observation. Following your suggestion, we have made it clear in Sec. 3.2 in the new revision.
>
> > Minor issue 2) In Figure 4, I cannot see the pattern the authors describe in "the lower right corner (corresponding to the labeled popular classes) of the heatmap is getting lighter and always lighter than the upper left corner." Maybe the Figure is not showing right or the matrices need to be renormalised to show the desired effect?
>
> We are sorry for the misunderstanding caused by our incorrect description and have corrected this description in the new revision. In the case of Fig. 4 with CADR's protocol, large class indexes indicate the popular classes and small class indexes indicate the rare classes. The lower left corner and upper right corner of the heatmap (i.e., *the class transitions between the popular classes and rare classes*) is getting lighter and always lighter than the upper left corner (i.e., *the class transitions among the popular classes*), which means the model is increasingly reluctant to transfer the class prediction to the rare classes during the pseudo-rectifying process, and the previous efforts to rectify the label to the rare class are forgotten by the model.
>
> ---
>
> Part (5/6)

---

> ### Author Response · Authors · 2022-11-16
> **Response to Reviewer buPA (4/6)**
>
> ### Next, we give our response to your questions:
>
> > Question 1) What exactly is $\xi$ computing? "Rectifying weight vector" is not a very precise description.
>
> Actually, we introduce $\xi$ only to formally express the pseudo-rectifying process, but we are sorry to bring you unnecessary misunderstanding. **Therefore, we have dropped $\xi$ in the new revision, and have rewritten the formula involved $\xi$ in a simpler and easier to understand way (please see Sec. 3.1).** The differences between the old and new versions are as follows:
>
> - **In the original version**, we use $\xi$ to formally indicate the change between the previous pseudo-label and current pseudo-label, i.e., denoting $x$ at epoch $e$ as $x^{e}$, the pseudo-rectifying process can be described as the change on $p$ by the next epoch: $p^{e+1}=\xi(\theta^{e},\theta^{e+1}) \circ p^{e}$, where $\xi$ is determined by the knowledge learned from the model parametrized by $\theta$ at epoch $e+1$. For example, given an unlabeled sample in CIFAR-10, the model outputs its pseudo-label $p^{e}=(0.1,0.5,0.4)$ at epoch $e$ and outputs its pseudo-label $p^{e+1}=(0.2,0.3,0.5)$ at epoch $e+1$. Then we can obtain $\xi=(2,0.6,1.24)$, i.e., $(0.1,0.5,0.4)\circ(2,0.6,1.24)=(0.2,0.3,0.5)$. However, after each update of model parameters, we cannot know the implicit value of $\xi$, even though it actually exists.
>
> - **In the new revision**, denoting $x$ at epoch $e$ as $x^{e}$, the pseudo-rectifying process can be described as the change on $p$ by the next epoch: $p^{e+1}=g\_{\theta}(p^{e})$, where $g\_{\theta}(p^{e})$ is a mapping from $p^{e}$ to $p^{e+1}$ determined by the knowledge learned from the model parametrized by $\theta$ at epoch $e+1$.
>
>
> > Question 2) In Equation (6), isn't $\sum^{k}\_{d=1}L\_{d}$ always equal to 1? So the first term on right-hand side of (6) is just the ratio of total number of predictions over the number of predictions that changed class (in a given batch), and is thus always larger than one?
>
> Actually, given $L\_{i}$ records the  number of class predictions belonging to class $i$ averaged on last $N\_{B}$ batches, $\sum^{k}\_{d=1}L\_{d}$ always equal to the batch size of unlablel data. But as you mentioned, the first term on right-hand side of Eq. (6) is  the ratio of total number of predictions over the number of predictions that changed class (in a given batch), and is thus always larger than one. This term controls the intensity of class transition at the batch-level. Please refer to our response to your *Weakness 1)* for more details and discussions.
>
> > Question 3) If $\mathbf{H}$ is a proper transition matrix, while do we need to renormalise $p$? Is it because of the $\xi$ term?
>
> We would like to clarify that the construction of  $\mathbf{H}$ is based on **class-level** statistics on **the unlabeled data**, *since its core role is to provide unbiased pseudo-rectifying guidance while avoiding the influence of biased labeled data*, i.e., its final effect will be presented at the class level. Therefore it will not be adapted to be used as a pseudo-label for each sample. Additionally, we provide experiments using $\mathbf{H}$ to serve as pseudo-label. As shown in Tab. d, the training almost collapse.
>
> Table d: The ablation studies on pseudo-label. We report the results (accuracy (%) / Geometric mean scores (GM)) on CIFAR-10 under CADR's protocol.
> |Method|$\gamma=20$ | $\gamma=50$ |  $\gamma=100$
> | :-----:| :-----:| :----: | :----: |
> |PRG w. $\mathbf{H}$ based pseudo-label| 23.66 / 17.33| 25.95 / 15.98| 28.35 / 21.30
> |PRG| 94.04 / 93.53 | 94.09 / 93.70 | 94.28 / 93.94 |
>
> ---
>
> Part (4/6)

---

> ### Author Response · Authors · 2022-11-16
> **Response to Reviewer buPA (3/6)**
>
> > Weakness 3) The random walk motivation is interesting but feels under-explored. Why take a simple step in the random walk? If we took $k$ steps, i.e., multiply by $\mathbf{H}^{k}$ instead of $\mathbf{H}$ in Equation (7), we could uncover more complex patterns of misclassification than simple pairwise class relations. For instance, if classes A and B are often misclassified as class C, but the model rarely classifies A as B and vice-versa, PRG would not capture this sort of relationship even though the transition matrix does.
>
> First of all, following your suggestion, we have conducted additional experiments on multiplying by $\mathbf{H}^{k}$ instead of $\mathbf{H}$, which is shown in Tab. b.
>
> Table b: The ablation studies on $\mathbf{H}^{k}$. We report accuracy (%) and Geometric mean scores (GM) on CIFAR-10 under CADR's protocol with $\gamma=20$.
>
> |$k$ | $1$ (default PRG) |$2$ | $5$ |  $10$ |
> | :-----:| :-----:| :----: | :----: | :----: |
> |Accuracy (%)| 94.04 | 91.33|87.76| 82.60 |
> |Geometric mean scores (GM)|93.53 |90.79| 85.80| 80.24 |
>
> We can observe that the performance is inversely proportional to $k$. The advantage of PRG is that $\mathbf{H}$ is updated in each iteration, which means that $\mathbf{H}$ is dynamically updated. As the model learns new knowledge, the past $\mathbf{H}$ may not be suitable for the pseudo-rectifying process anymore. If $\mathbf{H}^{k}$ is used, this means that we are using the same $\mathbf{H}$ multiple times for a given sample, which wastes the advantage of dynamic $\mathbf{H}$. We do not deny that perhaps $\mathbf{H}^{k}$ using a suitable $k$ or a dynamic selection of $k$ might yield better performance, but it is difficult for us to determine the value of $k$. Therefore, the PRG is designed for simplicity and exhibits superior the performance. Meanwhile, the above ablation experiments are shown in Tab. 6 in Sec. B of the new revision.
>
>
> > Weakness 4) Given that the motivation of the paper is not very grounded, the experiments seem not thorough enough since they are limited to CIFAR and mini-imagenet. CIFAR is notably noisy, which my benefit PRG. It would be interesting to see if the history of transitions is a useful artefact for general types of data other than images. Even simple and easy to run tabular datasets would already contribute to the overall message of the paper.
>
> Following your suggestion, we have conducted further experiments on **tabular MNIST (interpreting MNIST as a tabular data with 784 features)** by plugging PRG into **VIME** [1]. VIME is a prevailing self- and semi-supervised learning frameworks for tabular data with pretext task of estimating mask vectors from corrupted tabular data. We implement PRG above the semi-supervised learning component of VIME. PRG provides pseudo-rectifying guidance to rescale the pseudo-labels for  the original unlabeled sample in VIME. Specially, we replace the consistency loss used in VIME (mean squared error,Eq. (9) in [1]) with standard cross-entropy loss to makes PRG applicable to VIME. We use two protocols to show the performance advantage of PRG, including CADR's protocol and our protocol. As shown in Tab. c, except for CADR's protocol with $\gamma=20$ and our protocol with $n_{L}=40,N_{1}=10$ (it is worth noting that our upper limits of performance greatly exceed that of VIME), PRG outperforms original VIME by a large margin in the most of settings. The reason is that the scheme of class-transition-based pseudo-rectifying guidance is high-level and general (not limited to image data), which ultimately yields robust effect on the MNAR problem with tabular data. Meanwhile, the above additional experiments are shown in Tab. 11 in Sec. D.2 of the new revision. In a future version, we will explore further how PRG performs better on tabular data.
>
> Table c: Accuracy (%) on tabular MNIST. $\gamma$ is varied for CADR's protocol whereas $n\_{L}$ and $N\_{1}$ are varied for our protocol. Mean±Std are computed over 50 runs.
>
>
> |Method|$\gamma=20$ | $\gamma=50$ |  $\gamma=100$ | $n\_{L}=40,N\_{1}=10$|$n\_{L}=40,N\_{1}=20$|$n\_{L}=250,N\_{1}=100$|$n\_{L}=250,N\_{1}=200$|
> | :-----:| :-----:| :----: | :----: | :----: | :----: | :----: | :----: |
> |VIME| 63.38±4.42| 63.75±6.10| 64.80±2.76|50.13±7.56|30.73±8.69|60.58±2.68 | 21.44±0.58
> |+PRG| 59.41±14.45| 65.92±13.9 | 66.60±12.58 | 49.28±11.09 | 34.08±16.05 |  66.14±11.88 |  24.51±9.56
>
> ---
>
> Part (3/6)

---

> ### Author Response · Authors · 2022-11-16
> **Response to Reviewer buPA (2/6)**
>
> Continued
>
> ---
>
> Denoting the logit outputted from the model as $o$ (implying $p=\mathrm{Softmax}(o)$), with no gradient on pseudo-label $\tilde{p}$, we obtain
>     $$\frac{\partial \mathcal{L}\_{U}}{\partial o\_{c}}=-\sum^{k}\_{i}\frac{\tilde{p}\_{i}}{p\_{i}}\frac{\partial p\_{i}}{\partial o\_{c}}=-(\tilde{p}\_{c}-\tilde{p}\_{c}p\_{c}-\sum^{k}\_{i\neq c}\tilde{p}\_{i}p\_{c})=\sum^{k}\_{i}\tilde{p}\_{i}p\_{c}-\tilde{p}\_{c}=\frac{\sum^{k}\_{d=1}C\_{\hat{p}d}}{\mathcal{Z}\sum^{k}\_{d=1}\sum^{k}\_{d'=1}C\_{dd'} }\left(1-\frac{H\_{\hat{p}c}}{\mathcal{Z}\frac{L\_{c} }{\sum^{k}\_{d=1}L\_{d} }}\right)p\_{c}. $$
>     The larger the difference between $H\_{\hat{p}c}$ and $\frac{L\_{c} }{\sum^{k}\_{d=1}L\_{d} }$, the larger the gradient; Correspondingly, the smaller the difference between $H\_{\hat{p}c}$ and $\frac{L\_{c} }{\sum^{k}\_{d=1}L\_{d} }$, the smaller the gradient ($\frac{\partial \mathcal{L}\_{U}}{\partial o\_{c}}=0$ when $\frac{H\_{\hat{p}c}}{\mathcal{Z}\frac{L\_{c} }{\sum^{k}\_{d=1}L\_{d} }}=1$). This means that we intend to provide unbiased guidance *(because this is derived from the unlabeled data)* for the learning of unlabeled samples from the class level, so as to resist the influence of biased labeled samples. Meanwhile, if there are too many class transitions occur on the whole, $\frac{\sum^{k}\_{d=1}C\_{\hat{p}d}}{\sum^{k}\_{d=1}\sum^{k}\_{d'=1}C\_{dd'} }$ will decrease, which results in decreasing of $\mathcal{L}\_{U}$, i.e., suppress the trend of class transition overheating. The excessive class transitions in the self-training loop could yield confused supervision information that is not conducive to learning.
>
>
> Meanwhile, the ablation studies and discussion on the re-weighting are added into Sec. B in the new revision.
>
> > Weakness 2) The authors say $\dots$  but that does not necessarily require a **history of transitions** nor does it attenuate **"over-learning the samples of the labeled popular classes"**. I am just stating that it is not obvious (and thus potentially interesting) and the paper in its current version does not explain it very clearly.
>
> 1. For the utilization of the **history of transitions**, we would like to clarify that the usage of transition history is an exploration for addressing MNAR, i.e., what we want to express is that we propose a solution for MNAR: the usage of transition history. We find an interesting phenomenon in Fig. 4: class transitions occur intensively between the popular classes, and class transitions between the rare classes gradually disappear. For specific (*the following description is a modified version according to your minor issues 2*), the lower left corner and upper right corner of the heatmap (i.e., the class transitions between the popular classes and rare classes) is getting lighter and always lighter than the upper left corner (i.e., the class transitions among the popular classes), which means the model is increasingly reluctant to transfer the class prediction to the rare classes during the pseudo-rectifying process, and the previous efforts to rectify the label to the rare class are forgotten by the model.
>
>     At the beginning of training, the model also tries to generate class transitions from popular classes to rare classes. This historical information is captured by the PRG and used to guide the pseudo-rectifying process in the later stages of training. As analyzed above, the original FixMatch will no longer attempt to produce class transitions from popular classes to rare classes in the late training period. On the contrary, PRG can slow down the process of weakening this class transitions, i.e., *our intuition could be regarded as perturbations on some confident class predictions to preserve the pseudo-rectifying ability of the model with the usage of historical information.* More theoretical explanation can be found in our response to Weakness 1).
>
> 2. For the **attenuating of over-learning of the samples belonging to the labeled popular classes**. The model is misled by the baised labeled data (including popular classes of course) whereas PRG can suppress this trend thanks to the unbaised guidance fetched from the unlabled data. Please see our response to Weakness 1) for details.
>
> ---
>
> Part (2/6)

---

> ### Author Response · Authors · 2022-11-16
> **Response to Reviewer buPA (1/6)**
>
> ### Thank you for your detailed and constructive comments! Below is our response to your comments.
> ---
>
> ### First, we give our response to the weaknesses you mentioned:
>
> > Weakness 1) The definition of the transition matrix $\mathbf{H}$ is not very well motivated. It is not entire clear to me why Equation (6) is preferable to simply estimating $\mathbf{H}$ from the transitions observed in a batch of data. At least that would be a simple and telling baseline to compare against that, I believe, would add to the paper.
>
> 1. First of all, we have provided the additional ablation studies on the construction of $\mathbf{H}'$ by Eq. (6). Specifically, we simply estimating $\mathbf{H}$ from the transitions observed in the training and then use obtained $\mathbf{H}$ to construct PRG, whose results are shown in Tab. a.
>
>     Table a: The ablation studies on re-weighting by Eq. (6). We report the results (accuracy (%) / Geometric mean scores (GM)) on CIFAR-10 under CADR's protocol.
>
>     |Method|$\gamma=20$ | $\gamma=50$ |  $\gamma=100$
>     | :-----:| :-----:| :----: | :----: |
>     |PRG wo. re-weighting by Eq. (6)| 88.97 / 87.37| 91.73 / 91.28| 92.72 / 92.55
>     |PRG| 94.04 / 93.53 | 94.09 / 93.70 | 94.28 / 93.94 |
>
>     As shown in Tab. a, the re-weighting scheme can effectively boost the performance of PRG in MNAR because it controls the intensity of class transition together. The first term on the right-hand side of Eq. (6) (i.e., $\frac{\sum^{k}\_{d=1}L\_{d}}{\sum^{k}\_{d=1}\sum^{k}\_{d'=1}C\_{dd'}}$) rescales $H\_{ij}$ at the *batch-level* while the second term (i.e., $\frac{\sum^{k}\_{d=1}C\_{id}}{L\_{j}}$) rescales $H\_{ij}$ at the *class-level*. For $\frac{\sum^{k}\_{d=1}L\_{d}}{\sum^{k}\_{d=1}\sum^{k}\_{d'=1}C\_{dd'}}$, $\sum^{k}\_{d=1}L\_{d}$ is the total number of class predictions averaged on last $N\_{B}$ *batches* (according to *batch-level*) and $\sum^{k}\_{d=1}\sum^{k}\_{d'=1}C\_{dd'}$ is the total frequence of class transitions averaged on last $N\_{B}$ *batches* (also according to *batch-level*). This term suppress the excessive overall class transitions in the self-training loop, which could yield confused supervision information that is not conducive to learning. On the other hand, for $\frac{\sum^{k}\_{d=1}C\_{id}}{L\_{j}}$, $\sum^{k}\_{d=1}C\_{id}$ is the frequence of class predictions derived from class $j$ (according to *class-level*) and $L\_{j}$ is the number of class predictions belonging to class $i$ (also according to *class-level*). This term suppress the excessive class transitions towards classes that have been assigned many pseudo-labels.
>
> 2. In addtion, for our re-weighting of $\mathbf{H}$ in Eq. (6), we provide insight analysis based on the following theoretical justification. Overall, we give an explanation from the perspective of gradient. *Our re-weighting scheme potentially scales the gradient magnitude on the learning of the unlabeled data to mitigate adverse effects of biased labeled data, and suppresses the gradient magnitude when lass transition is overheated.* Letting $p$ be the naive soft label vector, $C\_{ij}$ be the batch-level frequency of class transitions that occur from class $i$ to class $j$, $L\_{i}$ be the  number of class predictions belonging to class $i$ averaged on last batches, by Eq. (6), we re-weight $\mathbf{H}$ by $H\_{ij}'= \frac{\sum^{k}\_{d=1}L\_{d}}{\sum^{k}\_{d=1}\sum^{k}\_{d'=1}C\_{dd'}} \times \frac{\sum^{k}\_{d=1}C\_{id}}{L\_{j}}\times H\_{ij}$ and obtain the rescaled pseudo-label vector $\tilde{p}^{\mathtt{}}=\mathrm{Normalize}(\mathbf{H}_{\hat{p}}'\circ p)$. Hence, the cross-entropy between prediction $p$ and $\tilde{p}^{\mathtt{}}$ can be formalized as
>     $$\mathcal{L}\_{U}=-\sum\_{c}^{k}\tilde{p}^{\mathtt{}}\log{p\_{c}}=-\sum\_{c}^{k}\left(\frac{\frac{\sum^{k}\_{d=1}L\_{d}}{\sum^{k}\_{d=1}\sum^{k}\_{d'=1}C\_{dd'}} \times \frac{\sum^{k}\_{d=1}C\_{\hat{p}d}}{L\_{c}}\times H\_{\hat{p}c}\times p\_{c}}{\mathcal{Z}}\right)\log{p\_{c}}=-\frac{\sum^{k}\_{d=1}C\_{\hat{p}d}}{\mathcal{Z}\sum^{k}\_{d=1}\sum^{k}\_{d'=1}C\_{dd'} }\sum\_{c}^{k}\left(\frac{H\_{\hat{p}c}\times p\_{c}}{\mathcal{Z}\frac{L\_{c} }{\sum^{k}\_{d=1}L\_{d} }}\right)\log{p\_{c}},$$
>     where $\mathcal{Z}$ is the normalize factor. $\frac{L\_{c} }{\sum^{k}\_{d=1}L\_{d} }$ can be regarded as the ratio of pseudo-labels belonging to class $c$ to all labels and $\frac{\sum^{k}\_{d=1}C\_{\hat{p}d}}{\sum^{k}\_{d=1}\sum^{k}\_{d'=1}C\_{dd'} }$ can be regarded as the ratio of class transitions derived from class $\hat{p}$ to the population transitions.
>
> ---
>
> To be continued
>
> Part (1/6)

---

### Official Review · Reviewer_nu14 · 2022-10-29

**Confidence:** 3
**Correctness:** 3
**Technical Novelty And Significance:** 3
**Empirical Novelty And Significance:** 3
**Recommendation:** 6

**Clarity, Quality, Novelty And Reproducibility:**

I think is paper is clearly written with a strong motivation, novel ideas, and sufficient reproducibility.

**Strength And Weaknesses:**

Strengths
1. This paper studies an important and practical problem in semi-supervised learning, which has been not studied much.
2. This paper is written very well. The motivation of research is presented clearly along with the limitation of existing approaches.
3. It is good to represent an existing approach (Sohn et al., 2020) as the framework of PRG and explain why it fails at MNAR scenarios.

Weaknesses
1. The paper claims to solve the MNAR problem, but I think it solves only one part of the problem: when the distribution of labeled data is balanced. How well does PRG perform in the opposite setting, i.e., balanced labeled but imbalanced unlabeled data?
2. The intuitions for Equations (6) and (7) are insufficient, although they are the core techniques of the proposed approach. Why do we need to use H_{ij}’ instead of H_{ij}? Why do we need the L vector to reweight the transition matrix?
3. I have some concerns on the experiments. Please see the following:
    1. I think the main competitor of PRG is CADR, which studies the same MNAR problem. However, the performance of CADR is presented only at Table 1 and missing from Table 2 and 3, and Figure 6. Why?
    2. In Table 1, the improvement of PRG from CADR is not always significant. In CIFAR-100 with \gamma=50, the accuracy of PRG is lower than that of CADR. When \gamma=200, the improvement is negligible considering the standard deviation. I don’t expect PRT to beat CDPR in all cases, but more explanations and analysis on the result would be helpful.


**Summary Of The Paper:**

A practical and challenging scenario called label Missing Not At Random (MNAR) is usually ignored in previous works on semi-supervised learning (SSL). In MNAR, the labeled and unlabeled data fall into different class distributions resulting in biased label imputation, which deteriorates the performance of SSL models. In this work, class transition tracking based Pseudo-Rectifying Guidance (PRG) is devised for MNAR. PRG unifies the history information of each class transition to activate the model’s enthusiasm for neglected classes, so as the quality of pseudo-labels on both popular classes and rare classes in MNAR could be improved.

**Summary Of The Review:**

I liked reading this paper, and I think it proposes good ideas to an important problem. My main concerns are that (a) this paper solves only a part of the MNAR problem, and (b) the accuracy improvement is not as strong as the authors claim in the paper, and there are not sufficient analysis and explanation on the result, i.e., Table 1.

---

> ### Author Response · Authors · 2022-11-16
> **Response to Reviewer nu14 (3/3)**
>
> 2. (Continued) Denoting the logit outputted from the model as $o$ (implying $p=\mathrm{Softmax}(o)$), with no gradient on pseudo-label $\tilde{p}$, we obtain
>     $$\frac{\partial \mathcal{L}\_{U}}{\partial o\_{c}}=-\sum^{k}\_{i}\frac{\tilde{p}\_{i}}{p\_{i}}\frac{\partial p\_{i}}{\partial o\_{c}}=-(\tilde{p}\_{c}-\tilde{p}\_{c}p\_{c}-\sum^{k}\_{i\neq c}\tilde{p}\_{i}p\_{c})=\sum^{k}\_{i}\tilde{p}\_{i}p\_{c}-\tilde{p}\_{c}=\frac{\sum^{k}\_{d=1}C\_{\hat{p}d}}{\mathcal{Z}\sum^{k}\_{d=1}\sum^{k}\_{d'=1}C\_{dd'} }\left(1-\frac{H\_{\hat{p}c}}{\mathcal{Z}\frac{L\_{c} }{\sum^{k}\_{d=1}L\_{d} }}\right)p\_{c}. $$
>     The larger the difference between $H\_{\hat{p}c}$ and $\frac{L\_{c} }{\sum^{k}\_{d=1}L\_{d} }$, the larger the gradient; Correspondingly, the smaller the difference between $H\_{\hat{p}c}$ and $\frac{L\_{c} }{\sum^{k}\_{d=1}L\_{d} }$, the smaller the gradient ($\frac{\partial \mathcal{L}\_{U}}{\partial o\_{c}}=0$ when $\frac{H\_{\hat{p}c}}{\mathcal{Z}\frac{L\_{c} }{\sum^{k}\_{d=1}L\_{d} }}=1$). This means that we intend to provide unbiased guidance *(because this is derived from the unlabeled data)* for the learning of unlabeled samples from the class level, so as to resist the influence of biased labeled samples. Meanwhile, if there are too many class transitions occur on the whole, $\frac{\sum^{k}\_{d=1}C\_{\hat{p}d}}{\sum^{k}\_{d=1}\sum^{k}\_{d'=1}C\_{dd'} }$ will decrease, which results in decreasing of $\mathcal{L}\_{U}$, i.e., suppress the trend of class transition overheating. The excessive class transitions in the self-training loop could yield confused supervision information that is not conducive to learning.
>
> 3. To sum up, without the re-weighting scheme on $\mathbf{H}$ and the usage of $L$, pure PRG can not work well at the **class level** (because there is no participation of estimated **class distribution**, i.e., $\frac{L\_{c} }{\sum^{k}\_{d=1}L\_{d} }$), which is exactly what we are trying to do to fight against MNAR (after all, the trouble MNAR brings us is the **mismatch of class distribution**). In sum, the motivation for the re-weighting on $\mathbf{H}$ is that we encourage the model to follow the guidance provided by $\mathbf{H}$ from the **class-level** (i.e., $\frac{H\_{\hat{p}c}}{\mathcal{Z}\frac{L\_{c}}{\sum^{k}\_{d=1}L\_{d}}}$ yields the effect of adjusting the class distribution of pseudo-labels), which enables $\mathbf{H}$ to play a better role in enhancing the model to preserve a certain probability to generate class transitions based on transition histroy (also corresponds to our initial motivation: providing pseudo-rectifying guidance at the **class level** to address MNAR).
>
>
> In addtion, the ablation studies and discussion on the re-weighting are added into Sec. B in the new revision.
>
>
> > Question 3) I think the main competitor of PRG is CADR, which studies the same MNAR problem. However, the performance of CADR is presented only at Table 1 and missing from Table 2 and 3, and Figure 6. Why?
>
> Following your suggesion, we have added comparisons with CADR in Tabs. 2 and 3, and Fig. 6 to address your concerns. Please refer to our new revision.
>
> > Question 4) In Table 1, the improvement of PRG from CADR is not always significant. In CIFAR-100 with $\gamma=50$, the accuracy of PRG is lower than that of CADR. When $\gamma=200$, the improvement is negligible considering the standard deviation. I don’t expect PRT to beat CDPR in all cases, but more explanations and analysis on the result would be helpful
>
> Thank you for your helpful suggestion! We note that original FixMatch's accuracy with 2500 labeled data (balanced distribution) on CIFAR-100 is 71.36%, which is almost the same order of magnitude as the number of labels used in CADR's protocol ($\gamma=50,\gamma=100$ and $\gamma=200$ respectively correspond to 1317, 2229 and 3875 total labels). Therefore we can infer that for the MNAR solution under  CADR's protocol, the upper limit of performance on CIFAR-100 is also around 71.36%. Therefore, it may be relatively difficult to continue improving on the high performance already achieved with CADR. However, PRG achieves a competitive performance compared to CADR on mini-ImageNet, which also consists of 100 classes, demonstrating the potential of our approach in MNAR.
>
>
> ## References
>
> [1] Xinting Hu, Yulei Niu, Chunyan Miao, Xian-Sheng Hua, and Hanwang Zhang. On non-random missing labels in semi-supervised learning. In International Conference on Learning Representations, 2022.
>
> ---
>
> Part (3/3)
>
> Thank you for your comments and we would be happy to discuss further with you!

---

> ### Author Response · Authors · 2022-11-16
> **Response to Reviewer nu14 (2/3)**
>
> > Question 2) The intuitions for Equations (6) and (7) are insufficient, although they are the core techniques of the proposed approach. Why do we need to use $\mathbf{H}\_{ij}'$ instead of $\mathbf{H}\_{ij}$? Why do we need the $L$ vector to reweight the transition matrix?
>
> Thank you for your comments. Please see item1 for intuition-based motivation, item 2 for theoretical analysis, and item 3 for concluding statements.
>
> 1. First, for our re-weighting scheme of $\mathbf{H}$ and the usage of $L$, we would like to provide an explanation from the perspective of intuition. The first term on the right-hand side of Eq. (6) (i.e., $\frac{\sum^{k}\_{d=1}L\_{d}}{\sum^{k}\_{d=1}\sum^{k}\_{d'=1}C\_{dd'}}$) rescales $H\_{ij}$ at the *batch-level* while the second term (i.e., $\frac{\sum^{k}\_{d=1}C\_{id}}{L\_{j}}$) rescales $H\_{ij}$ at the *class-level*. On the one hand, for $\frac{\sum^{k}\_{d=1}L\_{d}}{\sum^{k}\_{d=1}\sum^{k}\_{d'=1}C\_{dd'}}$, $\sum^{k}\_{d=1}L\_{d}$ is the total number of class predictions averaged on last $N\_{B}$ *batches* (according to *batch-level*) and $\sum^{k}\_{d=1}\sum^{k}\_{d'=1}C\_{dd'}$ is the total frequence of class transitions averaged on last $N\_{B}$ *batches* (also according to *batch-level*). This term suppresses the excessive overall class transitions in the self-training loop, which could yield confused supervision information that is not conducive to learning. On the other hand, for $\frac{\sum^{k}\_{d=1}C\_{id}}{L\_{j}}$, $\sum^{k}\_{d=1}C\_{id}$ is the frequence of class predictions derived from class $j$ (according to *class-level*) and $L\_{j}$ is the number of class predictions belonging to class $i$ (also according to *class-level*). This term suppresses the excessive class transitions towards classes that have been assigned overmuch pseudo-labels (caused by biased label imputation due to MNAR).
>
> 2. Next, we provide insight analysis based on the following theoretical justification. Overall, we give an explanation from the perspective of gradient. *Our re-weighting scheme potentially scales the gradient magnitude on the learning of the unlabeled data from **class level** to mitigate adverse effects of biased labeled data, and suppresses the gradient magnitude when lass transition is overheated.* Letting $p$ be the naive soft label vector, $C\_{ij}$ be the batch-level frequency of class transitions that occur from class $i$ to class $j$, $L\_{i}$ be the  number of class predictions belonging to class $i$ averaged on last batches, by Eq. (6), we re-weight $\mathbf{H}$ by $H\_{ij}'= \frac{\sum^{k}\_{d=1}L\_{d}}{\sum^{k}\_{d=1}\sum^{k}\_{d'=1}C\_{dd'}} \times \frac{\sum^{k}\_{d=1}C\_{id}}{L\_{j}}\times H\_{ij}$ and obtain the rescaled pseudo-label vector $\tilde{p}^{\mathtt{}}=\mathrm{Normalize}(\mathbf{H}_{\hat{p}}'\circ p)$. Hence, the cross-entropy between prediction $p$ and $\tilde{p}^{\mathtt{}}$ can be formalized as
>     $$\mathcal{L}\_{U}=-\sum\_{c}^{k}\tilde{p}^{\mathtt{}}\log{p\_{c}}=-\sum\_{c}^{k}\left(\frac{\frac{\sum^{k}\_{d=1}L\_{d}}{\sum^{k}\_{d=1}\sum^{k}\_{d'=1}C\_{dd'}} \times \frac{\sum^{k}\_{d=1}C\_{\hat{p}d}}{L\_{c}}\times H\_{\hat{p}c}\times p\_{c}}{\mathcal{Z}}\right)\log{p\_{c}}=-\frac{\sum^{k}\_{d=1}C\_{\hat{p}d}}{\mathcal{Z}\sum^{k}\_{d=1}\sum^{k}\_{d'=1}C\_{dd'} }\sum\_{c}^{k}\left(\frac{H\_{\hat{p}c}\times p\_{c}}{\mathcal{Z}\frac{L\_{c} }{\sum^{k}\_{d=1}L\_{d} }}\right)\log{p\_{c}},$$
>     where $\mathcal{Z}$ is the normalize factor. $\frac{L\_{c} }{\sum^{k}\_{d=1}L\_{d} }$ can be regarded as the ratio of pseudo-labels belonging to class $c$ to all labels and $\frac{\sum^{k}\_{d=1}C\_{\hat{p}d}}{\sum^{k}\_{d=1}\sum^{k}\_{d'=1}C\_{dd'} }$ can be regarded as the ratio of class transitions derived from class $\hat{p}$ to the population transitions.
>
> ---
>
> To be continued
>
> Part (2/3)

---

> ### Author Response · Authors · 2022-11-16
> **Response to Reviewer nu14 (1/3)**
>
> ### Thank you for your insightful feedback! Below is our response to your comments.
> ---
> > Question 1) The paper claims to solve the MNAR problem, but I think it solves only one part of the problem: when the distribution of labeled data is balanced. How well does PRG perform in the opposite setting, i.e., balanced labeled but imbalanced unlabeled data?
>
> First, we would like to clarify that our setting of MNAR does not include only the mentioned scenario. More precisely, we performed experiments on the extensive MNAR settings including
> -  Balanced labeled data and imbalanced unlabeled data (mentioned by reviewer) in CADR's protocol, our protocol and DARP's protocol with $\gamma\_u=1$.
> -  Imbalanced and mismatched labeled data / unlabeled data distributed with the same order in DARP's protocol. The results are shown in Fig. 6(b) with $\gamma\_u=50$ and $\gamma\_u=100$.
> -  Imbalanced and mismatched labeled data / unlabeled data distributed with the reversed order in our protocol. The results are shown in Fig. 6(a) and Tab. 2. We adopt this setting because it is more challenging than balanced labeled but imbalanced unlabeled data, since the distribution of the unlabeled data is inversely.
>
> **However, to further address your concern, we have provided more experiments on the setting of balanced labeled with imbalanced unlabeled data, which is summarized in Tab. a (also can be found in Tab. 1o in Sec. D.2 of the new revision.).** For specific, we set $n\_{L}=40$ with balanced labeled data and set $\gamma\_{u}=50,100,200$ for imbalanced unlabeled data, i.e., the class-wise number of unlabeled data $M\_{i}=M\_{1}\times \gamma\_{u}^{-\frac{k-i}{k-1}}$, where $M\_{1}=5000$ in CIFAR-10.
>
> As shown in Tab. a, PRG  outperforms all baseline methods by a large margin (it is worth noting that the performance of CADR is even weaker than original FixMatch), proving the robustness of PRG in the case of balanced labeled data with imbalanced unlabeled data due to the unbiased guidance derived from the class transition history.
>
> Table a: Accuracy (%) on CIFAR-10 with our protocol. The labeled data is balanced and the unlabeled data is imbalanced. More discussions on the baseline methods CoMatch and distribution alignment (DA) can be found in Sec. C.1 in Appendix.
>
>
> |Method|$\gamma\_{u}=50$| $\gamma\_{u}=100$ | $\gamma\_{u}=200$ |
> | :-----:| :-----:| :----: | :----: |
> |CoMatch| 52.73|46.20|38.85
> |FixMatch| 57.54|54.82|50.67
> |+DA| 54.08|46.71|41.37
> |+CADR| 49.38|45.27|42.30
> |+PRG|62.43| 62.44| 58.23 |
>
>
> Moreover, we notice CADR [1] does not provide the results of the setting of balanced labeled but imbalanced unlabeled data in its original paper. But CADR's protocol can also produce imbalanced unlabeled data by setting $\gamma\_{u}\neq 1$ (likewise, $M\_{i}=M\_{1}\times \gamma\_{u}^{-\frac{k-i}{k-1}}$). Although the labeled data is not balanced, to further explore the performance advantage of PRG, we conduct experiments under this setting. Following the experimental setup in Fig. 4 in [1], we set $\gamma=50$ for the labeled data and $\gamma\_{u}=10,20$ for the unlabeled data, respectively. The results are shown in Tab. b where PRG outperforms CADR by a large margin, showing ours superior bias removal on label imputation.
>
> Table b: Accuracy (%) / Geometric mean scores (GM) on CIFAR-10 with CADR's protocol. The labeled data is balanced and unlabeled data is imbalanced.
>
>
> |Method|$\gamma\_{u}=10$| $\gamma\_{u}=20$ |
> | :-----:| :-----:| :----: |
> |FixMatch|64.32 / 29.88| 63.72 / 32.55 |
> |+CADR|67.81 / 64.80| 64.01 / 58.92|
> |+PRG|88.97 / 87.37| 91.73 / 91.27|
>
> ---
>
> Part (1/3)

---

### Official Review · Reviewer_YaM3 · 2022-11-02

**Confidence:** 4
**Correctness:** 3
**Technical Novelty And Significance:** 3
**Empirical Novelty And Significance:** 3
**Recommendation:** 6

**Clarity, Quality, Novelty And Reproducibility:**

The writing and figures in this paper should be improved. The novelty of the paper is quite well. The authors give an anonymous link of the code.

**Strength And Weaknesses:**

Strength:
1.	The paper explains the necessity of solving MNAR problem clearly.
2.	The paper proposes a novel method for MNAR. The idea of focusing on label transition is quite interesting.
3.	The paper shows extensive experiments and the experimental results in the synthetic class-imbalanced settings demonstrated the effectiveness of PRG.
Weakness:
1.	The paper is a little hard to understand, especially for Section 3.1. The meaning of the symbol \xi is not clearly explained, and this symbol seems to disappear in pseudo-code although it appears in the main body.
2.	Figure 3 is the most important picture in the article ,but it is confusing. Although the style of the figure is nice, it does not help me understand PRG. There is no need to place a black box on (1,1).


**Summary Of The Paper:**

The authors focus on a challenging scenario called MNAR where the labeled and unlabeled data fall into different class distributions, which is different from traditional semi-supervised learning. In order to address the problem, the authors introduce a new technique called Pseudo-Rectifying Guidance (PRG). Using Markov random walk to get the class-level guidance information, PRG fully utilizes the history of each class's transition and makes the model concentrate on neglected classes more, therefore improves the performance of previous SSL models.

**Summary Of The Review:**

I tend to accept this paper since the problem is meaningful and the method show plenty of novelty. The main weaknesses are writing and figures, which can be improved in order to help readers understand the idea more quickly.

---

> ### Author Response · Authors · 2022-11-16
> **Response to Reviewer YaM3**
>
> ### Thank you for your helpful feedback! We are sorry for our potentially confusing writing and here are our responses to your comments.
> ---
> > Weakness 1) The paper is a little hard to understand, especially for Section 3.1. The meaning of the symbol $\xi$ is not clearly explained, and this symbol seems to disappear in pseudo-code although it appears in the main body.
>
> Actually, we introduce $\xi$ only to formally indicate the pseudo-rectifying process, but we are sorry to bring you unnecessary misunderstanding. **Therefore, we have dropped $\xi$ in the new revision, and have rewritten the formula involved $\xi$ in a simpler and easier way (please see Sec. 3.1).** The differences between the old and new versions are as follows:
>
> - **In the original version**, we use $\xi$ to formally express the change between the previous pseudo-label and current pseudo-label, i.e., denoting $x$ at epoch $e$ as $x^{e}$, the pseudo-rectifying process can be described as the change on $p$ by the next epoch: $p^{e+1}=\xi(\theta^{e},\theta^{e+1}) \circ p^{e}$, where $\xi$ is determined by the knowledge learned from the model parametrized by $\theta$ at epoch $e+1$. For example, given an unlabeled sample in CIFAR-10, the model outputs its pseudo-label $p^{e}=(0.1,0.5,0.4)$ at epoch $e$ and outputs its pseudo-label $p^{e+1}=(0.2,0.3,0.5)$ at epoch $e+1$, respectively. Then we can obtain $\xi=(2,0.6,1.24)$, i.e., $(0.1,0.5,0.4)\circ(2,0.6,1.24)=(0.2,0.3,0.5)$. However, after each update of model parameters, we cannot know the implicit value of $\xi$, even though it actually exists.
>
> - **In the new revision**, denoting $x$ at epoch $e$ as $x^{e}$, the pseudo-rectifying process can be described as the change on $p$ by the next epoch: $p^{e+1}=g\_{\theta}(p^{e})$, where $g\_{\theta}(p^{e})$ is a mapping from $p^{e}$ to $p^{e+1}$ determined by the knowledge learned from the model parametrized by $\theta$ at epoch $e+1$.
>
> > Weakness 2) Figure 3 is the most important picture in the article ,but it is confusing. Although the style of the figure is nice, it does not help me understand PRG. There is no need to place a black box on (1,1).
>
> Thank you for your suggestions! Following your suggestion, we have updated Fig. 3 in the new revision to make it more clear.
>
> ---
> Thank you for reading our response. We would be happy to discuss any further concerns!

---

### Official Review · Reviewer_Kxam · 2022-11-05

**Confidence:** 3
**Correctness:** 3
**Technical Novelty And Significance:** 2
**Empirical Novelty And Significance:** 3
**Recommendation:** 6

**Clarity, Quality, Novelty And Reproducibility:**

Clarify:

The paper is clearly written and easy to understand.
Please add value bars for those heat map plots of the class transition tracking matrices.

Quality and Novelty:

I like this simple and intuitive idea and appreciate the numerical work. However, the methodological innovation is fair, and the paper lacks a theoretical justification of the proposed procedure.

Reproducibility:

The authors provide an anonymous link for downloading code. This is great!

**Strength And Weaknesses:**

Strength:

- I enjoy reading this paper. It does a great job illustrating why each step is designed in this way and what effect it has on the training process. This paper has no theory, but the authors managed to explain the rationale using numerical examples from many different aspects.

- The methodology is simple, intuitive, and easy to implement. The improvement on real data experiments is also quite significant.

- I agree with the authors that it is a strong assumption that the labeled and unlabeled data are from the same distribution. The setting of missing not at random and knowing little about the missing scheme is quite common in practice. I think the authors studied a meaningful problem, and I am glad to see that they provided a simple approach that works so effectively.


Weaknesses:

- The major weakness is that this approach lacks theoretical justification. Although the re-weighting scheme has a good explanation, we have no idea whether it will have issues in some cases. For example, it is unclear to me why the particular ${\bf H}$ should be the correct choice of weights. I was imagining that a monotone transformation of entries of ${\bf H}$ may also serve as weights, and some likelihood-ratio based weights are more natural choices. To answer this question, we need to at least have some basic theoretical understanding. For example, at least in some very simple settings (e.g., linear classification settings, squared loss in FitMatch), we wish to know what the algorithm does in each update.

- The methodological innovation is not super large. As mentioned in the paper, the data-set level pseudo-rectifying guidance has been proposed in the literature. Using a class-level pseudo-rectifying guidance is more or less the same kind of ideas (but I do appreciate the simplicity of this approach and how it nicely fits the missing-not-at-random and imbalanced class settings).




**Summary Of The Paper:**

This paper studies semi-supervised classification when the labels are not missing at random. In this scenario, the labeled and unlabeled data cannot be considered as from the same distribution, and the existing semi-supervised learning algorithms face issues such as over-learning on popular classes and ignoring rare classes. The authors introduced a modification of the pseudo-rectifying procedure by tracking the class transitions. The rationale is that if the transition happens frequently between classes then we would like the algorithm to "memorize" it and avoid being over-confident on the imputed labels. The authors illustrate this idea with careful explanations and a number of numerical examples. The improvement on real data experiments is also impressive.

**Summary Of The Review:**

This is a well-motivated and well-written paper. The approach is not super novel, but it seems to have a lot of practical values. I think this is an okay paper. Some theoretical justification will greatly strengthen this paper.

---

> ### Author Response · Authors · 2022-11-16
> **Response to Reviewer Kxam (2/2)**
>
> 2. (Continued) The larger the difference between $H\_{\hat{p}c}$ and $\frac{L\_{c} }{\sum^{k}\_{d=1}L\_{d} }$, the larger the gradient; Correspondingly, the smaller the difference between $H\_{\hat{p}c}$ and $\frac{L\_{c} }{\sum^{k}\_{d=1}L\_{d} }$, the smaller the gradient ($\frac{\partial \mathcal{L}\_{U}}{\partial o\_{c}}=0$ when $\frac{H\_{\hat{p}c}}{\mathcal{Z}\frac{L\_{c} }{\sum^{k}\_{d=1}L\_{d} }}=1$). This means that we intend to provide unbiased guidance *(because this is derived from the unlabeled data)* for the learning of unlabeled samples from the class level, so as to resist the influence of biased labeled samples. Without this re-weighting scheme, pure PRG can not work well at the **class level** (because there is no participation of estimated **class distribution**, i.e., $\frac{L\_{c} }{\sum^{k}\_{d=1}L\_{d} }$), which is exactly what we are trying to do to fight against MNAR (after all, the trouble MNAR brings us is the **mismatch of class distribution**). In sum, the motivation for the re-weighting on $\mathbf{H}$ is that we encourage the model to follow the guidance provided by $\mathbf{H}$ from the **class-level** (i.e., $\frac{H\_{\hat{p}c}}{\mathcal{Z}\frac{L\_{c}}{\sum^{k}\_{d=1}L\_{d}}}$ yields the effect of adjusting the class distribution of pseudo-labels), which enables $\mathbf{H}$ to play a better role in enhancing the model to preserve a certain probability to generate class transitions based on transition histroy (also corresponds to our initial motivation: providing pseudo-rectifying guidance at the **class level** to address MNAR). Meanwhile, if there are too many class transitions occur on the whole, $\frac{\sum^{k}\_{d=1}C\_{\hat{p}d}}{\sum^{k}\_{d=1}\sum^{k}\_{d'=1}C\_{dd'} }$ will decrease, which results in decreasing of $\mathcal{L}\_{U}$, i.e., suppress the trend of class transition overheating. The excessive class transitions in the self-training loop could yield confused supervision information that is not conducive to learning.
>
> In addition, the discussion on the re-weighting are added into Sec. B in the new revision.
>
> > Weakness 2) The methodological innovation is not super large $\dots$ (but I do appreciate the simplicity of this approach and how it nicely fits the missing-not-at-random and imbalanced class settings).
>
> We would like to clarify that the focus of our method innovation is not to perform pseudo-rectifying guidance from the class level, but to propose a scheme of pseudo-rectifying guidance based on class transition tracking, which is introduced to SSL in MNAR for the first time. As mentioned in Sec. 1, PRG enhances the model to preserve a certain probability to generate class transition to rare classes when assigning pseudo-labels. This form of probability is based on class transition history, which has never been explored in solving the SSL problem.
>
> > Weakness 3) Please add value bars for those heat map plots of the class transition tracking matrices.
>
> Thank you for your helpful suggestion. We have add value bars for heat map plots of the class transition tracking matrices in Fig. 4 in the revision.
>
> ## References
>
> [1] David Berthelot, Nicholas Carlini, Ian Goodfellow, Nicolas Papernot, Avital Oliver, and Colin A. Raffel. Mixmatch: A holistic approach to semi-supervised learning. In Advances in Neural Information Processing Systems, 2019.
>
> ---
>
> Part 2/2
>
> Thanks for reading! We are more than happy to have further discussions.

---

> > ### Comment · Reviewer_Kxam · 2022-12-09
> > **Thank you for the response**
> >
> > Thank you for the detailed response to my comments. I appreciate the clarification that the major contribution (or novelty) of this paper is the design of a PRG scheme using class transition tracking.
> >
> > The theoretical justification remains somewhat unclear. I had a hard time understanding the sentence "we intend to provide unbiased guidance (because this is derived from the unlabeled data)". Are you saying that the gradient equal to zero is regarded as "unbiased guidance"? Is there a reason for this?
> >
> > Your explanation focuses on the effect of $H$ on the gradient of the cross entropy between the old $p$ and new $p$ (after one round of scaling/normalization by $H$). This offers interesting insights, but it is still not a rigorous theoretical justification. (Perhaps even a rigorous theoretical justification of the original PRG is challenging? A review of the theoretical understanding of PRG will be useful.)

---

> > > ### Author Response · Authors · 2022-12-10
> > > **Response to Reviewer Kxam - Round 2**
> > >
> > > ### Thank you for your further response and we sincerely hope that our reply could address your concerns.
> > >
> > > ---
> > >
> > > We are sorry for our potentially misleading sentences. We mentioned in our rebuttal: *"The larger the difference between $H\_{\hat{p}c}$ and $\frac{L\_{c} }{\sum^{k}\_{d=1}L\_{d} }$, the larger the gradient; Correspondingly, the smaller the difference between $H\_{\hat{p}c}$ and $\frac{L\_{c} }{\sum^{k}\_{d=1}L\_{d} }$, the smaller the gradient ($\frac{\partial \mathcal{L}\_{U}}{\partial o\_{c}}=0$ when $\frac{H\_{\hat{p}c}}{\mathcal{Z}\frac{L\_{c} }{\sum^{k}\_{d=1}L\_{d} }}=1$)."* According to this, the ***class-level unbiased guidance*** refers to that we tend to close the distance between $H\_{\hat{p}c}$ and $\frac{L\_{c} }{\sum^{k}\_{d=1}L\_{d}}$. Since $\mathbf{H}$ is a class-level statistic based on unlabeled data (containing the class transition history of rare classes and popular classes), we claim that it is a relatively unbiased guidance (at least not directly drawing information from biased labeled data). For SSL, the original guidance information for model learning is often derived from labeled data, e.g., $\mathcal{L}\_{L}$ in Eq. (2) in our paper. In MNAR, biased labeled data would seriously mislead the model's preference for classes, and PRG uses $\mathbf{H}$ from the unlabeled data to correct this undesirable preference, i.e., in conjunction with mentioned above, by rescaling the gradient to correct the model's learning direction. In addtion, we have supplemented the new ablation experiments in **Tab. a** to demonstrate its effectiveness. Specifically, we simply estimating $\mathbf{H}$ from the transitions observed in the training and then use the obtained $\mathbf{H}$ to construct PRG.
> > >
> > > Table a: The ablation studies on re-weighting by Eq. (6). We report the results (accuracy (%) / Geometric mean scores (GM)) on CIFAR-10 under CADR's protocol.
> > >
> > > |Method|$\gamma=20$ | $\gamma=50$ |  $\gamma=100$
> > > | :-----:| :-----:| :----: | :----: |
> > > |FixMatch| 56.26 / 41.90| 65.61 / 53.61| 72.28 / 60.35 |
> > > |+ PRG wo. re-weighting by Eq. (6)| 88.97 / 87.37| 91.73 / 91.28| 92.72 / 92.55 |
> > > |+ PRG| 94.04 / 93.53 | 94.09 / 93.70 | 94.28 / 93.94 |
> > >
> > > As shown in Tab. a, the re-weighting scheme can effectively boost the performance of PRG in MNAR because it controls the intensity of class transition. Further, without the re-weighting, as we mentioned in previous rebuttal: *"pure PRG can not work well at the class level (because there is no participation of estimated class distribution, i.e., $\frac{L\_{c} }{\sum^{k}\_{d=1}L\_{d} }$), which is exactly what we are trying to do to fight against MNAR (after all, the trouble MNAR brings us is the mismatch of class distribution)."* Although PRG can still help the original FixMatch achieve performance gains with the help of $\mathbf{H}$, it is nowhere near as good as with the results with re-weighted $\mathbf{H}$.
> > >
> > > As per your suggestion, we have tried our best to make some justifications from a theoretical perspective as much as possible. However, we wish to clarify that we do not deny that the re-weighting scheme (and PRG) is more empirical than theoretical. Noting that the rigorous theoretical analysis in the current SSL community usually needs to be based on a number of assumptions, we are afraid to admit that giving a rigorous theoretical analysis is difficult due to the short time at this stage (only two days left to rebuttal). Similar to some popular and successful SSL methods (e.g., FixMatch [1]) that seem to be heuristic methods based on experimental observations, PRG is also driven by methodology and experiments. However, we firmly believe that your suggestions are very valuable and that a PRG based on theoretical guarantees will be more contributive. We will aim to provide more theoretical perspectives and reflections in the future, so that the class transition tracking technique can facilitate the development of SSL. Thank you again for your comments.
> > >
> > > ## References
> > >
> > > [1] Kihyuk Sohn, David Berthelot, Chun-Liang Li, Zizhao Zhang, Nicholas Carlini, Ekin D. Cubuk, Alex Kurakin, Han Zhang, and Colin Raffel. Fixmatch: Simplifying semi-supervised learning with consistency and confidence. In Advances in Neural Information Processing Systems, 2020.

---

> ### Author Response · Authors · 2022-11-16
> **Response to Reviewer Kxam (1/2)**
>
> ### Thank you for your insightful feedback! Below is our response to your comments.
> ---
> > Weakness 1) The major weakness is that this approach lacks theoretical justification $\dots$ For example, at least in some very simple settings (e.g., linear classification settings, squared loss in FitMatch), we wish to know what the algorithm does in each update.
>
> Thank you for your comments. Please see item 1 for the additional experiments you mentioned, item 2 for the further theoretical justification on the re-weighting scheme of $\mathbf{H}$.
>
> 1. Fisrt, following your suggesions, we have added more ablation experiments you mentioned on the re-weighting scheme, i.e., we use a monotone transformation of entries of $\mathbf{H}$ to serve as weights and use some likelihood-ratio based weights for re-weighting, whose results are shown in Tab. a and Tab. b respectively. For monotone transformation, we adopt the **sharpening operation** (in fact, it can also be used for softening by adjusting the hyper-parameter $T$) in MixMatch [1] for experiments, i.e.,
>     $\textrm{Sharpen}(H\_{i},T)\_{j}=H\_{ij}^{\frac{1}{T}}/\sum^{k}\_{d}H^{\frac{1}{T}}\_{id},$
>     where $p$ is some input categorical distribution and $T$ is a hyperparameter. For **likelihood-ratio based weights**, we re-weight $\mathbf{H}$ as $H\_{ij}'= \frac{H\_{ij}}{\sum^{k}\_{d=1,d\neq j}H\_{id}}\times H\_{ij}.$ In the two tables, we can witness that the default PRG achieve the bset performance consistently.
>
>     Table a: Accuracy (%) / Geometric mean scores (GM) on CIFAR-10 with CADR's protocol ($\gamma=20$) and sharpening operation in [1]. The default PRG achieve a result of 94.04% / 93.53.
>
>
>     |$T$|$0.25$ | $0.5$ |  $2$
>     | :-----:| :-----:| :----: | :----: |
>     |Accuracy (%)| 82.73| 90.38| 78.89 |
>     |Geometric mean scores (GM)| 76.04| 89.57| 75.71 |
>
>     Table b: Results (accuracy (%) / Geometric mean scores (GM)) on CIFAR-10 with CADR's protocol and likelihood-ratio based weights .
>
>
>     |Method|$\gamma=20$ | $\gamma=50$ |  $\gamma=100$
>     | :-----:| :-----:| :----: | :----: |
>     |PRG w. likelihood-ratio based weights| 89.48 / 88.24| 90.44 / 88.97| 91.62 / 90.21
>     |PRG| 94.04 / 93.53 | 94.09 / 93.70 | 94.28 / 93.94 |
>
> 2. Second, we provide insight analysis into the experimental results based on the following theoretical justification. Overall, we give an explanation from the perspective of gradient. *Our re-weighting scheme potentially scales the gradient magnitude on the learning of the unlabeled data from **class level** to mitigate adverse effects of biased labeled data, and suppresses the gradient magnitude when class transition is overheated.* Letting $p$ be the naive soft label vector, $C\_{ij}$ be the batch-level frequency of class transitions that occur from class $i$ to class $j$, $L\_{i}$ be the  number of class predictions belonging to class $i$ averaged on last batches, by Eq. (6), we re-weight $\mathbf{H}$ by $H\_{ij}'= \frac{\sum^{k}\_{d=1}L\_{d}}{\sum^{k}\_{d=1}\sum^{k}\_{d'=1}C\_{dd'}} \times \frac{\sum^{k}\_{d=1}C\_{id}}{L\_{j}}\times H\_{ij}$ and obtain the rescaled pseudo-label vector $\tilde{p}^{\mathtt{}}=\mathrm{Normalize}(\mathbf{H}_{\hat{p}}'\circ p)$. Hence, the cross-entropy between prediction $p$ and $\tilde{p}^{\mathtt{}}$ can be formalized as
>     $$\mathcal{L}\_{U}=-\sum\_{c}^{k}\tilde{p}^{\mathtt{}}\log{p\_{c}}=-\sum\_{c}^{k}\left(\frac{\frac{\sum^{k}\_{d=1}L\_{d}}{\sum^{k}\_{d=1}\sum^{k}\_{d'=1}C\_{dd'}} \times \frac{\sum^{k}\_{d=1}C\_{\hat{p}d}}{L\_{c}}\times H\_{\hat{p}c}\times p\_{c}}{\mathcal{Z}}\right)\log{p\_{c}}=-\frac{\sum^{k}\_{d=1}C\_{\hat{p}d}}{\mathcal{Z}\sum^{k}\_{d=1}\sum^{k}\_{d'=1}C\_{dd'} }\sum\_{c}^{k}\left(\frac{H\_{\hat{p}c}\times p\_{c}}{\mathcal{Z}\frac{L\_{c} }{\sum^{k}\_{d=1}L\_{d} }}\right)\log{p\_{c}},$$
>     where $\mathcal{Z}$ is the normalize factor. $\frac{L\_{c} }{\sum^{k}\_{d=1}L\_{d} }$ can be regarded as the ratio of pseudo-labels belonging to class $c$ to all labels and $\frac{\sum^{k}\_{d=1}C\_{\hat{p}d}}{\sum^{k}\_{d=1}\sum^{k}\_{d'=1}C\_{dd'} }$ can be regarded as the ratio of class transitions derived from class $\hat{p}$ to the population transitions. Denoting the logit outputted from the model as $o$ (implying $p=\mathrm{Softmax}(o)$), with no gradient on pseudo-label $\tilde{p}$, we obtain
>     $$\frac{\partial \mathcal{L}\_{U}}{\partial o\_{c}}=-\sum^{k}\_{i}\frac{\tilde{p}\_{i}}{p\_{i}}\frac{\partial p\_{i}}{\partial o\_{c}}=-(\tilde{p}\_{c}-\tilde{p}\_{c}p\_{c}-\sum^{k}\_{i\neq c}\tilde{p}\_{i}p\_{c})=\sum^{k}\_{i}\tilde{p}\_{i}p\_{c}-\tilde{p}\_{c}=\frac{\sum^{k}\_{d=1}C\_{\hat{p}d}}{\mathcal{Z}\sum^{k}\_{d=1}\sum^{k}\_{d'=1}C\_{dd'} }\left(1-\frac{H\_{\hat{p}c}}{\mathcal{Z}\frac{L\_{c} }{\sum^{k}\_{d=1}L\_{d} }}\right)p\_{c}. $$
>
> ---
>
> To be continued
>
> Part (1/2)

---

### Author Response · Authors · 2022-11-17
**General Response to All Reviewers**

Dear Reviewers,

We would like to thank all reviewers for their detailed and valuable comments. We have responded to each reviewer individually to address any comments. Meanwhile, the manuscript has been updated while the new changes are highlighted in blue, which can be summarized as follows:

- In Appendix B, we further explain the meaning of our re-weighting scheme on $\mathbf{H}$ in Eq. (6) and provide more theoretical justification. Furthermore, we provide additional ablation studies on this re-weighting scheme, which is shown in Tab. 5 and Tab .6 in Appendix B.

- We add a paragraph to discuss more scenarios of MNAR in Appendix D.2. Meanwhile, we conduct additional experiments to demonstrate the superior and robust performance of PRG in the setting of balanced labeled data with imbalanced unlabeled data, which in shown in Tab. 10 in Appendix D.2.

- We add more data type (e.g., tabular data) for evaluation to comprehensively explore the potential of pseudo-rectifying guidance provided by PRG, whose results are shown in Tab. 11 in Appendix D.2.

- we are trying our best to implement other results or analysis suggested by reviewers, e.g., the comparison with CADR in Tab. 2, Tab. 3 and Fig. 6.

- We refresh Fig.3 and Fig. 4 to make them more clear and comprehensible.

- We modify the use of some symbols (e.g., $\xi$) to make the text more fluent and understandable.

- We add various missing details and fix a few statements (e.g., the description of experimental observation in Fig. 4) throughout the paper.

The valuable suggestions of reviewers have greatly benefited our paper. We are happy to see some further discussions to address any of your concerns.

Thanks again.

---

### Decision · Program_Chairs · 2023-01-20

**Decision:**

Reject

**Justification For Why Not Higher Score:**

This submission has failed to find interested reviewers and excitement among reviewers. For instance, it has not been clear to the reviewers how much the idea pushes research forward. Attempted clarifications during authors' responses after reviews have made some reviewers more doubtful than they were before.


**Justification For Why Not Lower Score:**

N/A

**Metareview: Summary, Strengths And Weaknesses:**

An idea based on rectifying guidance is presented for dealing with non MAR missing labels. The depth of the approach and the theoretical justification have not been fully understood, or they have been considered partially missing. The submission did not allow reviewers to be certain about the amount of novelty. In the short reviewing process and with the matched reviewers, the submission was not fully appreciated, and some replies made reviewers question further points which could not be resolved this quickly. Yet, it has become clear that the authors are putting a create effort into a potentially interesting topic, and it is thought that this will lead to strong outcomes. Reproducibility is appreciated.

**Summary Of Ac-Reviewer Meeting:**

The committee had quite a consensus around the fact that, as presented, this is a borderline submission. It has not become clear how NMAR (as treated here) should be something to be excited about for the general audience. No member was fully convinced by the (amount of) theoretical justifications. The efforts of the authors to try to explain their work have been praised.